# Reversible histone glycation is associated with disease-related changes in chromatin architecture

Qingfei Zheng [1], Nathaniel D. Omans[1,2], Rachel Leicher[3,4], Adewola Osunsade [1,4], Albert S. Agustinus [1], Efrat Finkin-Groner[1], Hannah D'Ambrosio[1], Bo Liu[5], Sarat Chandarlapaty [5], Shixin Liu[3] & Yael David[1,4,6,7]

Cellular proteins continuously undergo non-enzymatic covalent modifications (NECMs) that accumulate under normal physiological conditions and are stimulated by changes in the cellular microenvironment. Glycation, the hallmark of diabetes, is a prevalent NECM associated with an array of pathologies. Histone proteins are particularly susceptible to NECMs due to their long half-lives and nucleophilic disordered tails that undergo extensive regulatory modifications; however, histone NECMs remain poorly understood. Here we perform a detailed analysis of histone glycation in vitro and in vivo and find it has global ramifications on histone enzymatic PTMs, the assembly and stability of nucleosomes, and chromatin architecture. Importantly, we identify a physiologic regulation mechanism, the enzyme DJ-1, which functions as a potent histone deglycase. Finally, we detect intense histone glycation and DJ-1 overexpression in breast cancer tumors. Collectively, our results suggest an additional mechanism for cellular metabolic damage through epigenetic perturbation, with implications in pathogenesis.

---

[1] Chemical Biology Program, Sloan Kettering Institute, Memorial Sloan Kettering Cancer Center, New York, NY 10065, USA. [2] Tri-Institutional Training Program in Computational Biology and Medicine, New York, NY 10065, USA. [3] Laboratory of Nanoscale Biophysics and Biochemistry, Rockefeller University, New York, NY 10065, USA. [4] Tri-institutional PhD Program in Chemical Biology, New York, NY 10065, USA. [5] Human Oncology & Pathogenesis Program, Memorial Sloan Kettering Cancer Center, New York, NY, USA. [6] Department of Pharmacology, Weill Cornell Medical College, New York, NY 10065, USA. [7] Department of Physiology, Biophysics and Systems Biology, Weill Cornell Medical College, New York, NY 10065, USA. Correspondence and requests for materials should be addressed to Y.D. (email: davidshy@mskcc.org)

Glycation is one of the most prevalent NECMs and is characterized by the condensation of the aldehyde form of monosaccharides (such as glucose and fructose) or glycolytic by-products (such as methylglyoxal, MGO) with reactive amino acid residues (mainly primary amines in lysines and guanidino groups in arginines) via the Maillard reaction, forming stable adducts (Fig. 1)[1,2]. The initial glycation adduct can further oxidize and rearrange to form a series of stable products, which can undergo additional chemical transformations including the ability to form cross-links, yielding species generally referred to as advanced glycation end-products (AGEs)[1,3]. In diabetes, AGEs are highly abundant on both extra- and intra-cellular proteins and serve as a primary diagnostic tool through the quantification of glycated hemoglobin in the blood (A1C)[4]. Oxidative stress due to increase in reactive oxygen species (ROS) enhances the formation of AGEs, which in turn increases the presence of ROS in a positive feedback loop termed glycoxidation[5]. This phenomenon is particularly severe in cancer cells, which unlike healthy cells, primarily rely on anaerobic glycolysis for energy production (also referred to as the 'Warburg effect'), resulting in high levels of ROS and reactive carbohydrate species such as MGO[6,7]. Indeed, MGO adducts were identified in

many physiological samples including aged tissues and cancer tumors[8,9]. Thus, it is not surprising that various cellular mechanisms, such as GLO-1 and carnosine, have evolved to prevent MGO accumulation[10]. Moreover, recent evidence suggests enzymatic reversibility of early glycation intermediates (Fig. 1), although there is no known correction mechanism for cross-linked AGEs[11,12].

The core histone proteins (H2A, H2B, H3 and H4), which spool eukaryotic DNA into a chromatin structure, have extremely long half-lives that can reach months in non-proliferating cells[13]. Each histone protein contains an unstructured N-terminal tail that extends away from the nucleosome core particle (NCP) and undergoes a variety of PTMs on its abundant lysine and arginine residues, including methylation, acetylation and ubiquitination by a range of chromatin effectors that can write, read and erase these modifications[14]. Through the integration of diverse cellular stimuli, histone PTMs play a crucial role in determining cell fate by establishing and maintaining the epigenetic landscape[15]. An early low-resolution analysis of glycation performed on histones extracted from diabetic mouse liver cells indicated an increase in AGE levels compared to histones extracted from healthy liver cells[16]. A recent in vitro analysis of histone glycation was

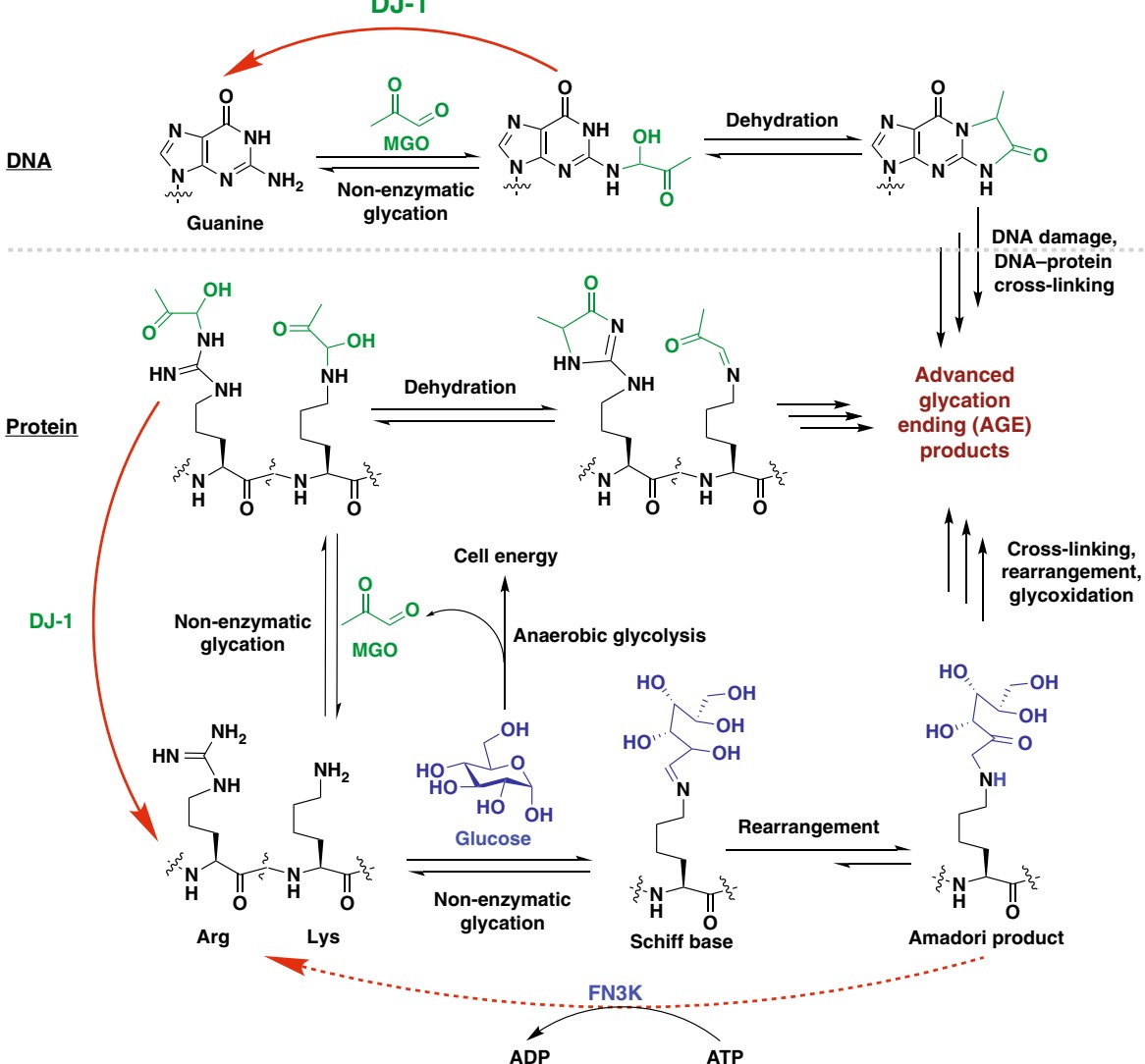

**Fig. 1** Protein and DNA glycation and deglycation cycle. Schematics of DNA (top) and protein (bottom) glycation by sugars (e.g. glucose) or glycolysis by-products (e.g. methylglyoxal) and deglycation by the enzymes DJ-1 and FN3K

performed using purified recombinant H2B and the linker histone H1 incubated with high levels of glucose and subjected to MS analysis. Several sites on both histones were found to be modified with various AGEs, including sites known to carry enzymatically added PTMs[17].

Here we perform a thorough analysis of the occurrence, mechanistic effect and pathological implications of histone MGO glycation in human cells. We characterize the inherent reactivity of all four core histones and identify H3 as the primary glycation substrate. We find that histone glycation disrupts assembly, stability and compaction of chromatin both in vitro and in cellulo. As a regulation mechanism, we identify the oncogenic protein DJ-1 to be a key histone deglycase that rescues glycation-induced damage. Finally, we show that breast cancer cells, xenografts, as well as patients' tumors have high basal histone glycation and DJ-1 levels. Together our results reveal the pathophysiological accumulation of histone glycation and identify an additional molecular mechanism linking metabolic perturbation with epigenetic misregulation in cancer.

## Results

**H3 is the prime target for MGO glycation**. MGO is an important glycolysis by-product, which was shown to modify proteins as well as DNA and has been implicated in contributing to cancer cell formation and metastasis[10,18]. To test the reactivity of MGO and chromatin components in vitro we applied a range of MGO concentrations corresponding to different sites:MGO ratios. These serve to both expedite the glycation reaction (that generates adducts accumulating over months to years in vivo) as well as mimic a variety of intermediates generated under different exposure conditions and times. We began by examining the inherent reactivity of each of the core histones towards MGO. For that, we incubated recombinant H2A, H2B, H3 or H4 with increasing amounts of MGO for 12 h, after which histones were analyzed by western blot using an anti-MGO antibody that detects these glycation adducts. Our results, presented in Fig. 2a, indicate that the hierarchy of histone reactivity with MGO is H3 > H4 > H2B > H2A, with H3 being highly reactive and H2A presenting only trace reactivity. It is noteworthy that glycation on H3 also compromises the anti-H3 epitope, causing a decrease in the overall anti-H3 signal in response to treatment with increased MGO concentration (Supplementary Figure 1, Supplementary Table 3).

To examine this selective reactivity in a more physiologically relevant context, we turned to reconstituted NCPs. Composed of recombinant wild-type histones and a minimal 147 bp '601' DNA fragment[19], NCPs represent the fundamental unit of chromatin. Since many of the sites of modification are hindered in NCPs compared to free histones, either by histone-histone or histone-DNA interaction, we predicted the glycation reaction will be slower. Reconstituted NCPs were thus incubated with higher ratios of MGO for 72 h, after which they were separated by SDS-PAGE and analyzed by western blot with anti-MGO. Indeed, this analysis revealed glycation adducts and rearranged cross-linked products forming in a concentration-dependent manner, where similar to the purified histones, H3 is the primary site of glycation (Fig. 2b, Supplementary Table 3). Finally, we turned to test the accumulation of MGO glycation adducts on chromatinized histones in live cells. The precise endogenous MGO concentration in healthy cells is unknown, and varies between cell types and metabolic states, however it was measured at 0.32 mM in cancer as well as cultured CHO cells[20,21]. Thus, we performed most of our cellular assays using 0.25–1 mM final MGO concentration in the media. To detect the formation of glycation on histones in living cells, we treated 293T cells with 0.5 mM MGO for 24 h. Subsequently, cells were harvested and the chromatinized histone

fraction was isolated by salt extraction[22]. Histones were analyzed by Coomassie Brilliant Blue (CBB) stain for purity and western blot for glycation content (Fig. 2c, Supplementary Table 3). Significant H3 and H4 glycation signal, as well as rearrangements and cross-linked products, occur in response to MGO treatment.

Although multiple cellular proteins can undergo glycation[23], we predicted that histone glycation has long-term effects due to the stable nature of histones. To test the accumulation of glycation on histones compared to other cellular proteins, we performed a pulse-chase experiment where 293T cells were treated with 1 mM MGO for 12 h followed by an MGO-free media chase of 12 and 24 h. Subsequently, cells were separated into soluble protein and chromatinized histone fractions, which were analyzed for glycated protein content. Our results show that while glycated soluble proteins are turned-over within one cell cycle, glycation on histones is retained even after 24 h of MGO-free media chase (Supplementary Figure 2).

In order to examine whether other histone PTMs are affected by these adducts, we cultured 293T cells in the presence of increasing amounts of MGO for 12 h, followed by histone extraction and western blot analysis with a panel of H3, H4 and H2B PTM-specific antibodies (Fig. 2d). Increased concentration MGO treatment produced an overall reduction in H3 and H4 PTMs signal, but not C-terminal ubiquitination on H2B. Our results thus indicate that chromatinized histone MGO glycation disrupts global histone enzymatic PTM levels.

**Histone glycation disrupts nucleosome assembly and stability.** Our in vitro and in cellulo results suggest that histones are modified with MGO and these modifications quickly rearrange to cross-linked products. We thus hypothesized that MGO adducts could have a significant effect on nucleosome stability. To test this, we performed a "one-pot" nucleosome assembly in the presence of unmodified or glycated histones (using the same conditions presented in Fig. 2a). To aid in detection of the assembled NCP we used a DNA fragment that contains a 5' biotin on a short PEG linker[19]. The nucleosome assemblies were analyzed by native gel electrophoresis and visualized using ethidium bromide, which intercalates the DNA. In parallel, a second native gel was transferred to a membrane and analyzed by western blot with anti-biotin. This western blot analysis only detects fully assembled nucleosomes since free DNA does not transfer to a membrane (Supplementary Figure 3). While unmodified histones or MGO-treated H2A and H2B do not seem to disrupt nucleosome assembly, H3 and H4 glycation had a detrimental effect on NCP formation, with a direct negative correlation between the level of glycation on each histone and the amount of stable nucleosome formed (Figs. 2a and 3a).

It was recently shown that DNA can undergo glycation by MGO, which induces the DNA damage response and cytotoxicity[24–26]. Since nucleosomal DNA is less accessible due to its intimate bonding with the histone octamer, we set out to examine whether it is modified by MGO in our setup. For that we treated NCPs with increasing concentrations of MGO for 12 h and used PCR amplification to detect the occurrence of glycation, since the canonical DNA polymerase does not amplify a compromised template. Indeed, our analysis detected MGO glycation on the nucleosomal DNA, as illustrated by the diminished signal in PCR amplification (Fig. 3b, Supplementary Table 3). Since glycation rearrangement can result in cross-linking, we next wondered whether the glycations we detected separately on DNA and histones could actually form a physical DNA-histone cross-link. To address this possibility, we treated the nucleosomes with high salt, which neutralizes non-covalent DNA-histone contacts and induces nucleosome disassembly, and then analyzed the

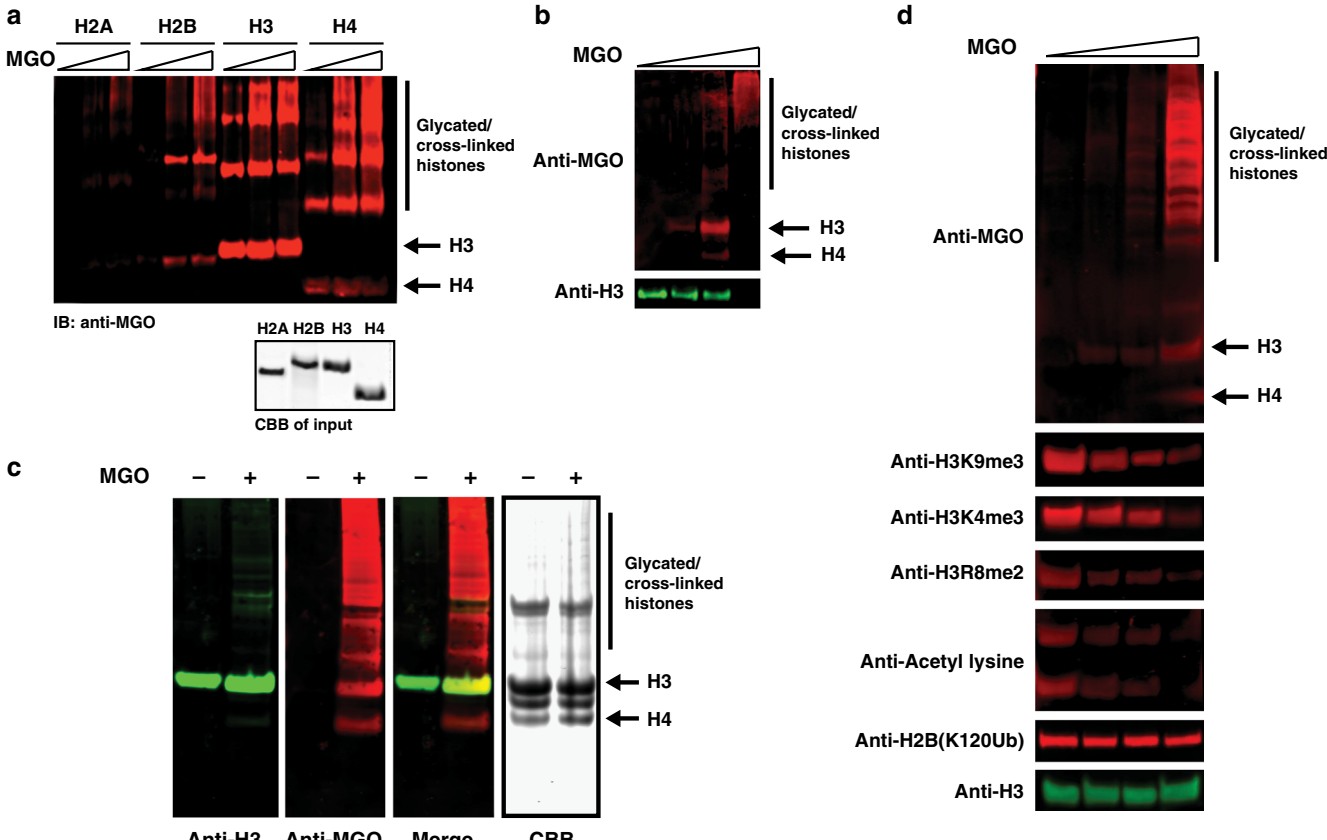

**Fig. 2** Histone H3 and H4 are prime targets for glycation in vitro and in cellulo. **a** Glycation of free histones. Recombinant purified histones (indicated in the coomassie brilliant blue stained SDS-PAGE) were incubated with increasing concentrations of MGO (0.1, 0.5 or 1 mM, corresponding to 4:1, 1:1 or 1:2 sites: MGO stoichiometry, respectively) for 12 h at 37 °C. Analysis was performed by SDS-PAGE followed by western blot with anti-MGO. **b** Reconstituted nucleosomes were treated with 0, 0.5, 5 or 100 mM of MGO (corresponding to 1:2, 1:20 or 1:400 sites:MGO stoichiometry, respectively) for 72 h at 37 °C, after which the nucleosomes were analyzed as described for free histones. **c** Histone glycation in 293T cells after 0.5 mM MGO treatment for 24 h. Histones were extracted by high salt and analyzed by coomassie or western blotted with the indicated antibodies. **d** 293T cells were incubated with 0, 0.25, 0.5 or 1 mM of MGO for 12 h at 37 °C, after which histones were extracted with high salt and analyzed by western blot with the indicated histone PTM antibodies

sample by native gel. Treating NCPs with low MGO concentrations induced glycation but not histone-DNA cross-linking, as the nucleosome signal disappears upon high salt treatment. However, high concentration MGO treatment resulted in robust DNA-histone cross-linking, as the complex remained intact after salt treatment (Fig. 3c). Together these results indicate that high concentrations of (or long exposure to) MGO can induce both histone-histone and histone-DNA cross-linking in chromatin, potentially having harmful effects on its dynamic nature.

**Histone glycation disrupts chromatin architecture.** Since we observed glycation-induced DNA-histone cross-linking on nucleosomes, we hypothesized that this can affect the architecture of chromatin on both a local and global scale. To examine this possibility, we utilized reconstituted nucleosomal arrays composed of 12 repeats of 601 DNA assembled in a similar manner to NCPs. These 12 NCP repeats represent the minimal chromosomal fragment shown to have a native local chromatin fold[27] and their compaction state can be evaluated by a variety of biophysical assays[28]. To first verify that nucleosomal arrays are stable in the presence of MGO, we characterized them before and after MGO treatment using agarose-polyacrylamide gel electrophoresis (APAGE) (Fig. 4a, Supplementary Table 3). To gain a quantitative measurement of NCP stability in an array context, we used a

single-molecule high-resolution optical tweezers assay[29]. This assay measures the precise force required to unravel each nucleosome in an array by tethering a single array between two beads – each held by a laser trap – and applying force to pull them apart. Performing this assay on untreated and MGO-treated arrays indicated that more force was applied to unravel nucleosomes in treated arrays, strongly suggesting that on a single nucleosome level MGO-treated arrays are more stable, presumably due to additional intra-nucleosomal bonding (Fig. 4b).

To test the relative compaction state of MGO-treated arrays, we carried out Mg$^{2+}$ precipitation assays, which takes advantage of the propensity of chromatin to compact and precipitate upon the addition of divalent metal ions. When arrays were treated with a low concentration of MGO for 12 h, Mg$^{2+}$ precipitation assays indicated the arrays were decompacted (Fig. 4c, green line), as more Mg$^{2+}$ was required to precipitate these compared to untreated arrays (red line). However, high concentration MGO treatment produced the exact opposite effect of array hyper-compaction (Fig. 4c, blue line and Fig. 4g, scheme). To verify that the higher concentration of MGO mimics a longer exposure we treated arrays with an even lower MGO concentration and tested their compaction state every 6 h. Indeed, initial time points indicated array decompaction, while later time points exhibited higher compaction, presumably due to rearranging and cross-linking (Fig. 4c). Similarly, when we used MNase digestion, which

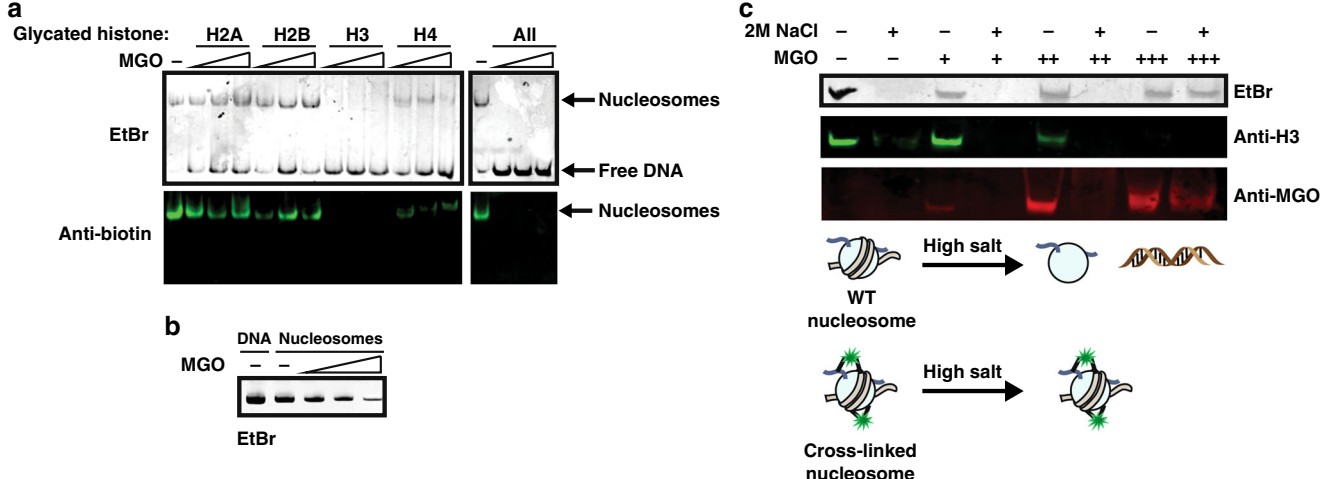

**Fig. 3** Histone glycation disrupts nucleosome assembly and stability. **a** Histones were treated as described in Fig. 2a before used in nucleosome assembly with salt dialysis in the presence of a strong nucleosome positioning DNA sequence ('601'). Samples were separated on a native gel that was either stained with intercalating agent (EtBr) (top) or transferred and western blotted with anti-biotin (bottom). **b** Nucleosomes were treated with 0, 2, 30 or 100 mM of MGO (corresponding to 1:8, 1:120 or 1:400 sites:MGO stoichiometry, respectively) for 12 h at 37 °C and then used as templates in a PCR reaction. Products were separated on a 1 % agarose gel and stained with EtBr. **c** Top: histone-DNA glycation mediated cross-linking was examined using the sensitivity of unmodified nucleosomes to high salt treatment. Nucleosomes were treated with 0, 0.5, 5 or 100 mM of MGO (corresponding to 1:2, 1:20 or 1:400 sites: MGO stoichiometry, respectively) for 72 h at 37 °C after which they were either untreated or treated with 2 M NaCl. Samples were then separated on a native gel and either imaged by EtBr or transferred and analyzed by western blot with the indicated antibodies. Bottom: Schematic representation of untreated versus 100 mM MGO-treated nucleosomes before and after high salt treatment

relies on the cleavage of accessible DNA, low-concentration MGO treatment of arrays yielded a higher cleavage rate compared to untreated arrays, while high-concentration MGO treatment yielded MNase protected arrays (Fig. 4d, g, scheme). Finally, to test whether MGO-histone adducts induce similar changes in chromatin architecture in living cells, we performed high-resolution and genome-wide DNA accessibility analysis. We treated 293T cells with 0.5 mM MGO for 12 h, followed by a global ATAC-seq analysis that indicated a significant MGO-induced abrogation in DNA accessibility, particularly in transcription start sites (Fig. 4e, f, and Supplementary Figure 4). Together, these results propose that histone glycation disrupts both local and global chromatin architecture by altering histone-DNA interactions.

**DJ-1 actively deglycates histones in vitro and in cellulo.** DJ-1 (also referred to as PARK7) has emerged in recent years as a protector of metabolism-associated cellular stress, reversing primarily ROS damage in neurons but also protein and DNA glycation under a variety of conditions[30]. To determine whether DJ-1 is a potential regulator of histone glycation, we first tested its ability to selectively bind a glycated H3 tail. We synthesized a peptide corresponding to the H3 N-terminal tail (residues 1–18) with a C-terminal biotin (Supplementary Figure 5) and subjected it to MGO treatment followed by incubation with recombinant DJ-1 (Supplementary Figure 6) and streptavidin pull down. In parallel, we incubated glycated or non-glycated peptide with 293T total cell lysate. After the pull down, the beads were washed, boiled, separated on SDS-PAGE and analyzed by western blot with anti-DJ-1 (Fig. 5a). In both cases, DJ-1 was enriched by glycated compared to the non-glycated peptide bait. Next, we tested the ability of DJ-1 to reverse histone glycation in vitro. We treated free full-length recombinant H3 with limiting amounts of MGO for 2 h and then added either wild-type or catalytically dead (C106A) recombinant DJ-1[31] to the reaction (Supplementary Figure 6). The reactions were then analyzed by SDS-PAGE

followed by a western blot analysis with anti-MGO and anti-H3. Our results show that wild type but not C106A mutant DJ-1 was able to remove the glycations and restore H3 (Fig. 5b and Supplementary Figure 7). Similar results were obtained when NCPs were treated with MGO and wild type or C106A mutant DJ-1 (Fig. 5c & Supplementary Figure 7).

To examine whether DJ-1 acts on histones in living cells, we transfected 293T cells with either wild type or C106A mutant DJ-1 (Supplementary Figure 8). After 24 hours, cells were treated with 0.5 mM MGO for an additional 12 h, harvested, and histones were extracted with high salt as described above. Glycation analysis revealed that cells transfected with wild-type DJ-1 presented no trace of histone MGO glycation, while cells expressing the catalytically dead DJ-1 showed high histone glycation levels, similar to non-transfected sample (Fig. 5d). To test whether DJ-1 can reverse MGO-driven changes in chromatin compaction state we observed before, we used $Mg^{2+}$ precipitation assays to assess the compaction of glycated arrays before and after DJ-1 treatment. While low concentration MGO treatment decompacts chromatin, as observed before (dark green line), wild-type DJ-1 (black line), but not C106A mutant DJ-1 (light green line), reverses this effect to a comparable level of the untreated arrays (red line) (Fig. 5e). Next, we utilized an shRNA construct to knock down the level of endogenous DJ-1 in 293T cells (Fig. 5f) and tested the sensitivity of these cells to MGO. DJ-1 knockdown cells exhibit basal histone glycation in the absence of MGO, higher sensitivity to MGO treatment (Fig. 6g) and lower viability (Supplementary Figures 9 and 10), all of which could be rescued by DJ-1 overexpression, further illustrating the key role it plays in regulating histone glycation in cells. Finally, to test whether these effects are correlated with the epigenetic changes we previously observed (Fig. 2d), we extracted histones from 293T cells where DJ-1 was overexpressed or downregulated and compared the levels of different histone modifications. Indeed, our results reveal that DJ-1 deficiency promotes even more significant MGO-induced epigenetic damage that can be rescued by wild-type DJ-1 overexpression and

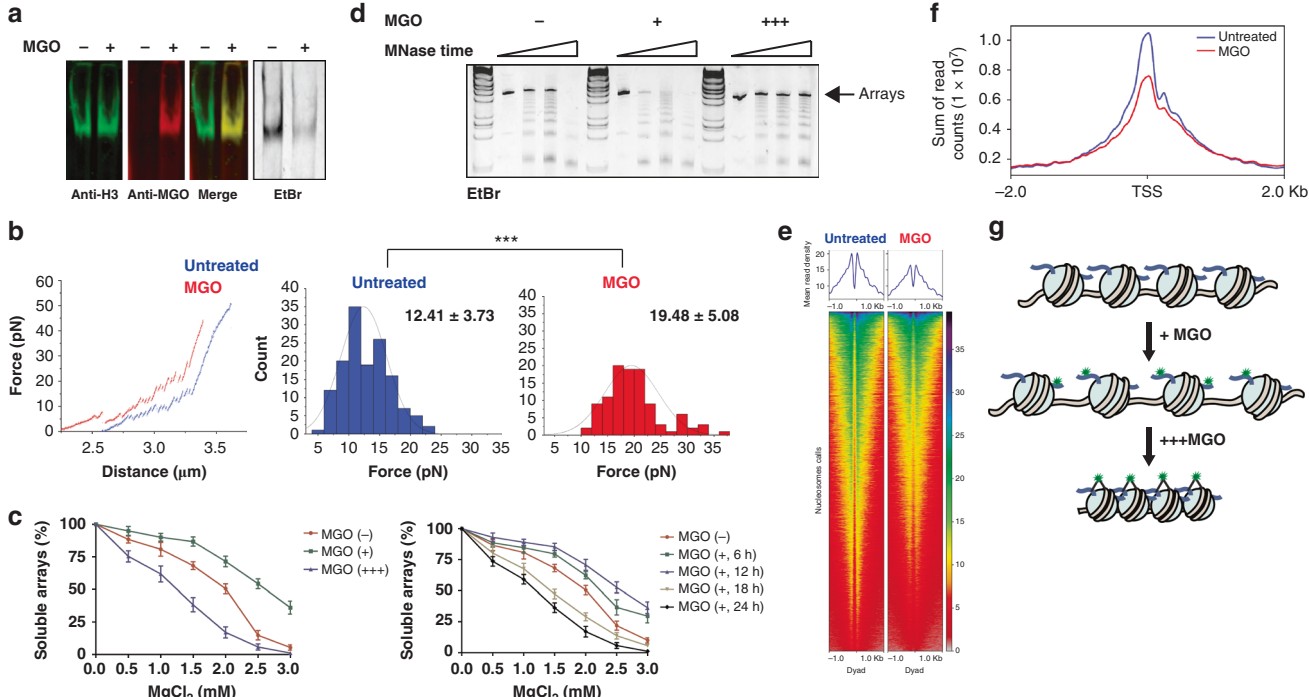

**Fig. 4** Histone glycation changes chromatin architecture in vitro and in cellulo. **a** Reconstituted nucleosomal 12mer arrays were incubated with 1 mM MGO for 12 h and analyzed by APAGE (agarose polyacrylamide native gel), either with EtBr or western blot utilizing the indicated antibodies to show the stability of the arrays upon MGO treatment. **b** Nucleosomal arrays were treated with 100 mM MGO (corresponding to 1:30 sites:MGO stoichiometry) for 12 h at 4 °C. Left: Representative single-molecule unfolding trajectories of MGO-treated (red) and untreated (blue) array obtained by an optical tweezer experiment. Right: Histograms of forces required to unravel each nucleosome in the MGO-treated nucleosomal array (red) and untreated (blue) array. Statistical analysis was based on sampling more than 50 single-molecule data and performing a *t* test. **c** Left: magnesium compaction assay of nucleosomal arrays treated with different concentrations of MGO. (+) 10 mM (corresponding to 1:3 sites:MGO stoichiometry); (+++) 100 mM MGO (corresponding to 1:30 sites:MGO stoichiometry). Right: array treated with low MGO concentration for different time points. (+) 2 mM MGO (corresponding to 1:0.6 sites:MGO stoichiometry). Error bars represent standard deviation from three different experiments. **d** MNase digestion of nucleosomal arrays treated with increasing amounts of MGO. Samples were separated on a native gel and stained with EtBr. **e** ATAC-seq performed on 293T cells either untreated or treated with 0.5 mM MGO for 12 h. Heatmaps showing the density of mapped ATAC-seq reads at consensus nucleosome peaks and 1000 bp up and downstream from the nucleosome dyad. Line graphs above show the same data as the mean read depth over all nucleosome peaks. **f** Line graph of the sum of ATAC-seq reads over RefSeq annotated transcriptional start sites show a decrease in chromatin accessibility in MGO-treated cells. **g** Schematic representation of the results presented in (**b**–**d**)

returned to normal state, but not by DJ-1 C106A inactive mutant (Supplementary Figure 10).

**Histones are basally glycated in breast cancer.** DJ-1 was recently defined as an oncogene due to its overexpression in many cancers, as well as the fact that its knockdown decreases cancer cell proliferation and induces their apoptosis[32] (Supplementary Figure 9). Since breast cancer was shown to have a particularly high metabolic rate and ROS levels, we set out to examine histone glycation and sensitivity to MGO in these cell types. We first tested DJ-1 expression in breast cancer cell lines and our results show that they all display significantly higher levels compared to 293T (Supplementary Figure 11). We next examined the basal histone glycation level in these cells as well as their sensitivity to MGO treatment. Our results, presented in Fig. 6a, indicate that all tested cell lines have both higher basal H3 glycation (compared to undetectable levels in 293T cells) as well as sensitivity to MGO treatment. When SKBR3 breast cancer cells were pre-treated with carnosine, an endogenously produced MGO scavenger, both basal glycation levels of histones and sensitivity to MGO treatment were dramatically reduced (Fig. 6b). Moreover, when DJ-1 was knocked down in SKBR3 cells using shRNA, the level of histone glycation significantly increased and the overall viability of the cells went down (Supplementary Figure 12).

Since MGO glycation disrupts histone acetylation (Fig. 2d), we hypothesized that these two modifications compete for the same sites. To test this, we pre-treated 293T cells with either the p300 histone acetyl-transferases (HAT) inhibitor curcumin or the histone deacetylases (HDAC) inhibitor SAHA. Following MGO treatment, histones were extracted and analyzed with anti-MGO and anti-acetyl-lysine. Indeed, cells pre-treated with HAT inhibitor showed lower levels of acetylation and high basal glycation as well as higher sensitivity to MGO, similar to breast cancer cells (Fig. 6c). Analogously, pre-treatment with HDAC inhibitors, which stabilizes histone acetylation, resulted in higher levels of histone acetylation and lowered sensitivity of histones to glycation (Fig. 6c).

To test the accumulation of histone glycation in a more physiological environment we obtained breast cancer xenograft tumors, which were extracted from mice kept under the same conditions. Following their removal from the animal, tumors were homogenized and total lysate was analyzed by western blot with anti-DJ-1. Our results indicate that these tumors display variable DJ-1 levels (Supplementary Figure 11). Analyzing histones extracted from these tumors revealed most of them contain high basal histone MGO glycation levels, further corroborating the data observed in cell culture (Fig. 6d). Notably, MGO glycation was not detected in the soluble protein fraction from the same sample (Supplementary Figure 11). Interestingly,

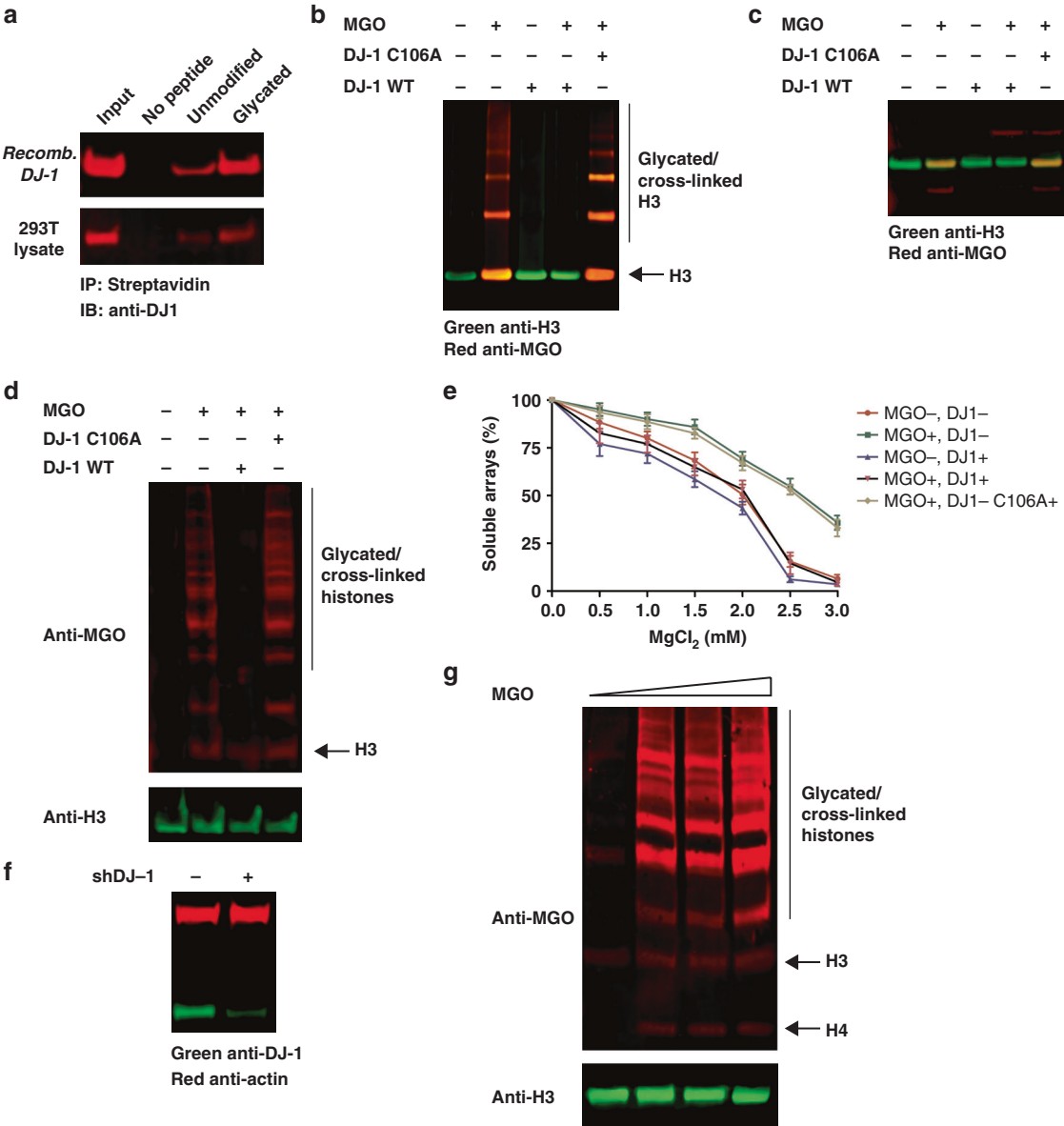

**Fig. 5** Histone glycation is repaired by DJ-1 in vitro and in cellulo. **a** Either non-glycated or glycated biotin-H3 tail (residues 1-18) was incubated with either recombinant DJ-1 or a 293T lysate at 4 °C for 2 h. Subsequently, peptides were enriched by streptavidin bead pull-down and analyzed by anti-DJ-1. **b** Full-length H3 was either treated or not treated with 1 mM of MGO (corresponding to 1:2 sites:MGO stoichiometry) for 2 h at 37 °C after which, either wild-type DJ-1 or catalytically inactive mutant C106A DJ-1 was added for an additional hour. Samples were analyzed on SDS-PAGE with the indicated antibodies. **c** NCPs were treated with MGO in the presence or absence of either wild-type DJ-1 or catalytically inactive mutant C106A DJ-1. Samples were analyzed by SDS-PAGE followed by a western blot with the indicated antibodies. **d** 293T cells were either not transfected, or transfected with wild-type or catalytically inactive mutant C106A DJ-1, and then grown in the absence or presence of 0.5 mM MGO for 12 h after which histones were extracted in a high salt buffer and analyzed by western blot with the indicated antibodies. **e** $Mg^{2+}$ compaction experiments of nucleosomal arrays treated with MGO in the presence or absence of wild-type DJ-1. Error bars represent the standard deviation from three different experiments. **f** shRNA of DJ-1. 293T cells were transfected with shRNA plasmid and 24 h later were harvested. Total cell lysate was analyzed by western blot with anti-DJ-1 and anti-Actin as a loading control. **g** 293T cells transfected with shRNA DJ-1 were incubated with 0, 0.25, 0.5 and 1 mM of MGO for 12 h at 37 °C, after which histones were extracted with high salt and analyzed by western blot with the indicated antibodies

the highest DJ-1 expressing cell line and tumor also exhibit the highest histone glycation levels (see discussion). Finally, we obtained 5 fresh-frozen breast cancer patients' tumor samples (T), each extracted side-by-side with a non-tumor (NT) sample. Tissue samples were subsequently homogenized and both soluble and chromatin fractions were analyzed as described above. Our analysis clearly shows high and specific histone glycation in the tumor over the non-tumor samples with a distinct band for H3 (Fig. 6e). Importantly, tumor samples showed a significantly

higher DJ-1 expression level compared to a non-tumor sample from the same patient (Supplementary Figure 11). In aggregate, these results illustrate that glycation specifically accumulates on histone proteins under endogenous conditions in a disease-state specific manner.

## Discussion
To date, little is known about histone glycation and how it affects chromatin structure and function. To study this phenomenon, we

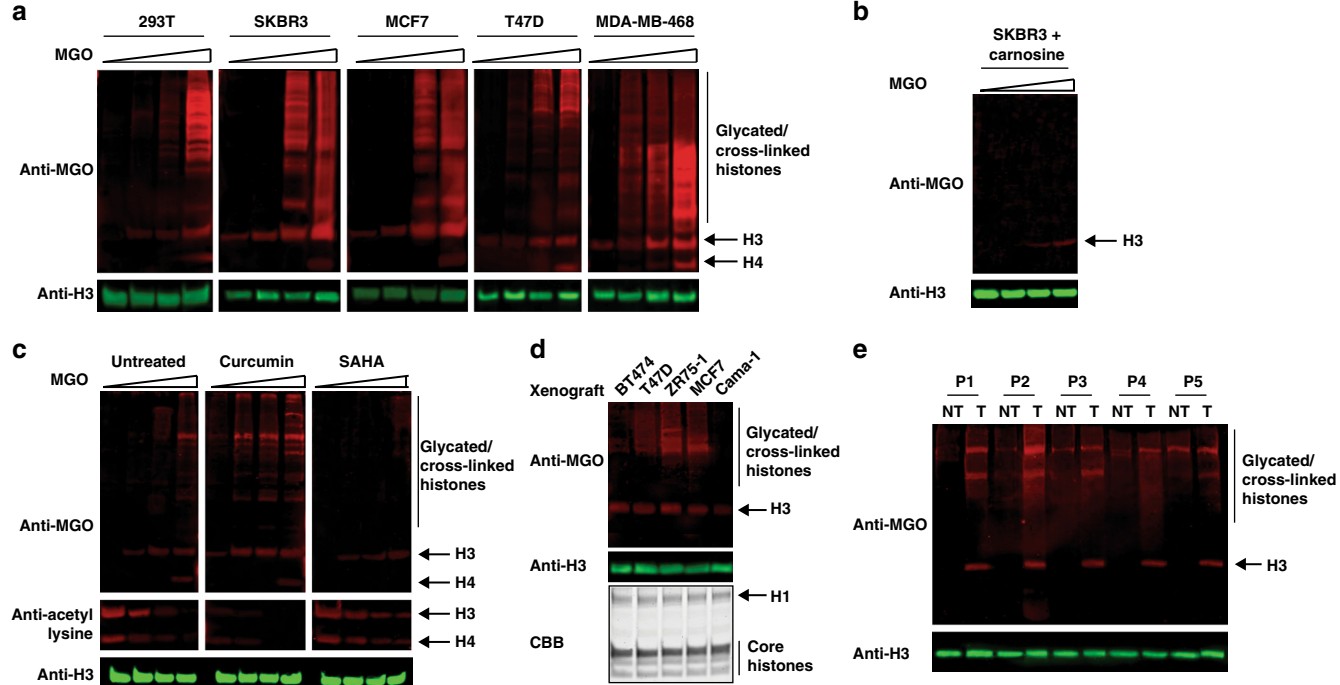

**Fig. 6** Breast cancer cells and tumors display high basal histone glycation and sensitivity to MGO. **a** Indicated breast cancer cell lines (and 293T as a reference) were untreated or treated with 0.25, 0.5 or 1 mM MGO for 12 h, after which histones were extracted by high salt and analyzed by western blot on the same SDS-PAGE with the indicated antibodies. **b** SKBR3 breast cancer cell line was pre-treated for 2 h with 10 mM carnosine, after which increasing amounts of MGO were added to the media. Histones were extracted as described above and samples were analyzed by western blot with anti-MGO. **c** 293T cells were pre-treated with histone acetyl-transferases (curcumin, 50 μM) or deacetylases (SAHA, 5 μM) inhibitors for 2 h after which increasing amounts of MGO were added to the media. Histones were extracted as described above and samples were analyzed by western blot with the indicated antibodies. **d** Indicated breast cancer xenografts tumors were analyzed for histone glycation, by extracting the histones with high salt and analyzing them by western blot with the indicated antibodies. Corresponding cytosolic glycation analysis and DJ-1 expression can be found in Supplementary Figure 11. **e** Non-tumor (NT) and tumor (T) samples were obtained from five different breast cancer patients (P1-P5) and both soluble and histone fractions were extracted and analyzed as described above. Corresponding DJ-1 expression levels are found in Supplementary Figure 11

performed a comprehensive analysis at both low and high resolutions utilizing a variety of methods examining the occurrence of as well as the local and global chromatin changes induced by histone glycation. Our results reveal that histone glycation occurs primarily on H3, induces major chromatin changes in vitro and in cellulo and natively accumulates in certain breast cancer tumors. Furthermore, we identify DJ-1 as a key enzyme in the rescue of these phenotypes by both protecting against and reversing histone glycation damage.

Glycation of proteins by MGO was reported to accumulate in aging cells as well as a variety of pathologies including diabetes and cancer[33]. As a non-enzymatic chemical modification, the appearance of glycated product is directly correlated with the concentration of the reactants and reaction time. Since histone proteins have some of the longest half-lives in the cell, they are highly susceptible to the accumulation of damage in vivo. This would occur primarily on the histones' highly accessible and lysine- and arginine-rich tails, which undergo a variety of enzymatic modifications and participate in the epigenetic regulation of transcription. Our results show that normal cells do not accumulate glycation adducts on histones (possibly through the activity of DJ-1, which is widely expressed), but robustly display them in response to a short MGO treatment (Fig. 2c, d). Under these conditions other cellular proteins do not accumulate significant glycation (Supplementary Figure 2). The MGO concentrations we used for most of these assays were relatively low, although possibly higher than its physiological concentration[21]. These were used to mimic long pathological exposures, and under these conditions histones indeed have similar levels of glycation

to untreated breast cancer cells, xenografts and patient tumors (Fig. 6a, d, e). Together, these results indicate that histone glycation is a pathophysiological outcome of an imbalanced cellular metabolic state, which can be mimicked in culture. While it is yet to be determined whether these glycation events participate in driving pathology, recent advances illustrate that similar carbonyl stress from alcohol consumption and high sugar diets induces non-enzymatic chromatin damage and DNA mutations, suggesting this is the case[34,35]. Moreover, disease implications of glycation are not limited to cancer, as other conditions are associated with perturbed metabolic states, such as diabetes, wherein protein glycation is a hallmark of disease progression. Since the epidemiological correlation between diabetes and other pathologies has no known mechanistic explanation, histone glycation could offer insight into these changes in cell fate through epigenetic regulation.

Metabolic changes induce rapid cellular responses ranging from the upregulation of signaling cascades to long-term changes in the transcriptional program[36]. However, our results indicate that there might be a direct effect of metabolic changes on the epigenetic landscape through H3 glycation, which occurs in proportion to the concentration of MGO in the cellular environment and alters histone PTM signature (Fig. 2d). It is important to note that since we used antibodies for PTM detection, it is possible that in some cases glycation events compromise the ability of the antibody to bind its epitope. However, since these antibodies recognize PTM-modified lysines and arginines, which are also the primary sites for MGO adducts, it seems more likely that MGO competes for these sites rather than occluding the

epitopes. Among the tested PTMs, H3R8me2s showed higher sensitive to MGO treatment than lysine methylation (H3K4 and H3K9), which corresponds with the higher reactivity of MGO towards arginine over lysine (Fig. 2a, d)[9,37]. H3/H4 acetylation seems to be the most sensitive to MGO treatment, which could be attributed to the high dynamics of histone acetylation, which was shown to be much more readily added and removed than methylation[38,39]. In fact, our results indicate that changing the stability of acetylation by perturbing its writers (HATs) or erasers (HDACs) directly alters histone sensitivity to glycation (Fig. 6c). These results suggest a general role for lysine acetylation in protecting histones from non-enzymatic damage.

Our biophysical assays address the propensity of nucleosomal arrays to change compaction state in response to glycation. We utilized a short chromatin fragment composed of 12 nucleosomal repeats, whose compaction state is easily recorded. Our results indicate that at lower MGO concentrations this fiber relaxes, while higher MGO concentrations induce fiber compaction. A plausible explanation for this phenomenon is embedded in the property of the chemical reaction between MGO and protein side chains. At low concentrations, MGO reacts quickly with arginines and lysines to cap the amino side chains' positive charges (Fig. 3a), in analogy to acetylation, thus weakening the electrostatic interactions with the negatively charged DNA, causing fiber relaxation. On the other hand, under higher MGO concentrations glycation occurs more rapidly and densely to induce histone-histone and/or histone-DNA cross-linking, inducing chromatin compaction. These results suggest two stages of histone glycation damage: a short-term stage that may cause a change in the transcriptional program by increasing the accessibility of certain genes, and long-term damage that changes the global dynamics of chromatin leading to a malfunction in cellular epigenetic regulation. Interestingly, our ATAC-sequencing data shows that cells treated with MGO exhibit a general chromatin compaction compared to untreated cells, suggesting that in vivo the transition between the two glycation stages occurs more rapidly. Changes in TSS accessibility could explain the previously reported rapid changes in cellular transcriptome in response to MGO treatment[40].

Cancer cells have upregulated metabolism, generating high MGO and ROS levels which in healthy cells would lead to apoptosis through protein- and DNA-associated stress responses. DJ-1 was shown by us and others to be upregulated in cancer. Despite its overexpression, we find that cancer cells nevertheless display high levels of histone glycation (Fig. 6, Supplementary Figures 9, 11 and 13 for proposed mechanism). We thus propose that these cells are dependent on DJ-1 for maintaining a lower level of stress in order to evade cell death. Indeed, shRNA experiments of DJ-1 in cancer cells induce rapid growth arrest and cell death[41]. In support, our viability assays show higher survival rates for DJ-1 overexpressing cells, even at high MGO treatment levels, and lower survival for DJ-1 knockdown cells (Supplementary Figures 9 and 12). Together with our observation of high level of DJ-1 in breast cancer tumors, these results support DJ-1 as an appealing potential therapeutic target for cancer treatment.

In summary, although age- and pathology-associated MGO damage was identified in physiological samples, its effect on histones and chromatin has not been determined yet. Here, we identify the accumulation of glycation specifically on histones in breast cancer cell lines, xenografts and patient tumors, but not in normal cultured cells. We determined that the molecular ramifications of histone glycation are disruption of chromatin structure and function, which potentially contributes to genomic instability and cancer progression, providing an additional link between metabolic state and cell fate. Finally, we propose a

therapeutic avenue, targeting DJ-1 as a key gatekeeper of cancer cell survival.

In line with our findings, during the review process, Marnett and colleagues have published a metabolomics mass spectrometry analysis detecting lysine and arginine MGO glycation on histones in cells and tissues, sensitive to DJ-1 expression[42].

## Methods

**General Materials and Methods.** Amino acid derivatives and coupling reagents were purchased from AGTC Bioproducts. Dimethylformamide (DMF), dichloromethane (DCM), triisopropylsilane (TIS) were purchased from Fisher Scientific and used without further purification. Tris(2-carboxyethyl)phosphine hydrochloride (TCEP) was purchased from Thermo Scientific. 2-(7-aza-1H-benzotriazole-1-yl)-1,1,3,3-tetramethyluronium hexafluorophosphate (HATU) was purchased from Oakwood Chemicals, and O-(benzotriazol-1-yl)-N,N,N′,N′-tetramethyluronium hexafluorophosphate (HBTU) was purchased from Fisher Scientific. Trifluoroacetic acid (TFA) was purchased from Fisher Scientific. N,N-diisopropylethylamine (DIPEA) was purchased from Fisher Scientific. Analytical reversed-phase HPLC (RP-HPLC) was performed on an Agilent 1200 series instrument with an Agilent C18 column (5 μm, 4 × 150 mm), employing 0.1% TFA in water (HPLC solvent A), and 90% acetonitrile, 0.1% TFA in water (HPLC solvent B), as the mobile phases. Analytical gradients were 0–70% HPLC buffer B over 30 min at a flow rate of 0.5 mL/min, unless stated otherwise. Preparative scale purifications were conducted on an Agilent LC system. An Agilent C18 preparative column (15–20 μm, 20 × 250 mm) or a semi-preparative column (12 μm, 10 mm × 250 mm) was employed at a flow rate of 20 mL/min or 4 mL/min, respectively. HPLC Electrospray ionization MS (HPLC-ESI-MS) analysis was performed on an Agilent 6120 Quadrupole LC/MS spectrometer (Agilent Technologies). UV spectrometry was performed on NanoDrop 2000c (Thermo Scientific). Biochemicals and media were purchased from Fisher Scientific or Sigma-Aldrich Corporation unless otherwise stated. T4 DNA ligase, DNA polymerase and restriction enzymes were obtained from New England BioLabs. Primer synthesis and DNA sequencing were performed by Integrated DNA Technologies and Genewiz, respectively. PCR amplifications were performed on a Bio-Rad T100™ Thermal Cycler. Centrifugal filtration units were purchased from Sartorius, and MINI dialysis units purchased from Pierce. Size exclusion chromatography was performed on an AKTA FPLC system from GE Healthcare equipped with a P-920 pump and UPC-900 monitor. Sephacryl S-200 columns were obtained from GE Healthcare. All the western blots were performed using the primary antibodies annotated in Supplementary Table 1 and fluorophore-labeled secondary antibodies annotated in Supplementary Table 2 following the protocol recommended by the manufacture. Blots were imaged on Odyssey CLx Imaging System (Li-Cor).

**Peptide synthesis.** Standard Fmoc-based Solid Phase Peptide Synthesis (Fmoc-SPPS) was used for the synthesis of peptides in this study. The peptides were synthesized on ChemMatrix resins with Rink Amide to generate C-terminal amides. Peptides were synthesized using manual addition of the reagents (using a stream of dry $N_2$ to agitate the reaction mixture). For amino acid coupling, 5 eq. Fmoc protected amino acid were pre-activated with 4.9 eq. HBTU, 5 eq. HOBt and 10 eq. DIPEA in DMF and then reacted with the N-terminally deprotected peptidyl resin. The resin was washed with DMF and reaction completion was monitored using the Kaiser test[43]. Fmoc deprotection was performed in an excess of 20% (v/v) piperidine in DMF, and the deprotected peptidyl resin was washed thoroughly with DMF to remove trace piperidine. Cleavage from the resin and side-chain deprotection were performed with 95 % TFA, 2.5% TIS and 2.5% $H_2O$ at room temperature for 1.5–2 h. The peptide was then precipitated with cold diethyl ether, isolated by centrifugation and dissolved in water with 0.1 % TFA followed by RP-HPLC and ESI-MS analyses. Preparative RP-HPLC was used to purify the peptide of interest.

**Recombinant histone expression and purification.** Recombinant human histones H2A, H2B, H3, and H4 were expressed in E. coli BL21 (DE3), and purified by flash reverse chromatography as previously described[44]. The purified histones were analyzed by RP-LC-ESI-MS: H2A $[M + H]^+$ observed, 13964.85 Da; expected 13964.3 Da; H2B $[M + H]^+$ observed, 13758.79 Da; expected, 13758.9 Da; H3.2 $[M + H]^+$ observed, 15256.95 Da; expected, 15256.8 Da; H4 $[M + H]^+$ observed, 11236.80 Da; expected, 11236.1 Da.

**Octamer and '601' preparation.** Octamers were prepared as previously described[44]. Briefly, recombinant histones were dissolved in unfolding buffer (20 mM Tris-HCl, 6 M GdmCl, 0.5 mM DTT, pH 7.5), and combined with the following stoichiometry: 1.1 eq. H2A, 1.1 eq. H2B, 1 eq. H3.2, 1 eq. H4. The combined histone solution was adjusted to 1 mg/mL concentration transferred to a dialysis cassette with a 7000 Da molecular cutoff. Octamers were assembled by dialysis at 4 °C against 3 × 1 L of octamer refolding buffer (10 mM Tris-HCl, 2 M NaCl, 0.5 mM EDTA, 1 mM DTT, pH 7.5) and subsequently purified by size exclusion chromatography on a Superdex S200 10/300 column. Fractions containing octamers were combined, concentrated, diluted with glycerol to a final 50% v/v and

stored at −20 °C. The 147-bp 601 DNA fragment was prepared by digestion from a plasmid containing 30 copies of the desired sequence (flanked by blunt EcoRV sites on either site), and purified by PEG-6000 precipitation as described before[45].

**Mononucleosome assembly with wild-type histone octamers**. The mononucleosome assembly was performed according to the previously described salt dilution method with slight modification[19]. Briefly, the purified wild-type octamers were mixed together with 601 DNA (1:1 ratio) in a 2 M salt solution (10 mM Tris pH 7.5, 2 M NaCl, 1 mM EDTA, 1 mM DTT). After incubation at 37 °C for 15 min, the mixture was gradually diluted (9 × 15 min) at 30 °C by dilution buffer (10 mM Tris pH 7.5, 10 mM NaCl, 1 mM EDTA, 1 mM DTT). The assembled mononucleosomes were concentrated and characterized by native gel electrophoresis (5% acrylamide gel, 0.5 × TBE, 120 V, 40 min) using ethidium bromide staining.

**Mononucleosome assembly with glycated histones**. To test glycated histone assembly into mononucleosomes (Fig. 3a), a one-pot assembly assay was performed as previously described with slight modifications[46]. Briefly, the four histones (H2A, H2B, H3.2 and H4) were first glycated by the corresponding concentrations of MGO at 37 °C for 12 h (shown in Fig. 2a), and then mixed together with 601 DNA in equal molar ratio (at 3 μM concentration) in PBS buffer (pH 7.4) containing 2 M NaCl and incubated at room temperature for 1–2 h. The mixture was then diluted with PBS to 0.25 M NaCl concentration and incubated at room temperature for another hour. The assemblies were analyzed by native page electrophoresis (5% acrylamide gel, 0.5 × TBE, 120 V, 40 min), followed by ethidium bromide staining or transferred to a PVDF membrane and analyzed by western blot.

**Nucleosomal array assembly with wild-type histone octamers**. Dodecameric repeats of the 601 sequence separated by 30-bp linkers were produced from pWM530 using EcoRV digestion and PEG-6000 precipitation according to the published procedure[47]. Homotypic dodecameric arrays were assembled from purified octamers and recombinant DNA in the presence of buffer DNA (MMTV) by salt gradient dialysis as previously described[6]. The resulting arrays were purified and concentrated using Mg$^{2+}$ precipitation at 4 °C[28].

**Expression of recombinant wild-type DJ-1 and C106A mutant**. The pET3a-His-DJ1 plasmid was a kind gift from Michael J Fox Foundation MJFF (Addgene plasmid # 51488). The His$_6$-tagged DJ-1 C106A mutation was cloned by inverse PCR using pET3a-His-DJ1 as the template and the following primer sequences: 5′-GCCGCCATTGCCGCAGGCCCGACCGC-3′ and 5′-AATCAGGCCTTTGC GGTTCTCCTGCTCTTTCAGG-3′. The His$_6$-tagged wild-type DJ-1 and DJ-1 C106A mutation was expressed in E. coli Rosetta (DE3) cells with an overnight IPTG induction at 16 °C. The bacterial pellet was lysed by sonication and lysate cleared by centrifugation at 12,000 r.p.m. for 30 min. Lysate was loaded on Ni$^{2+}$-NTA agarose (GE Healthcare) affinity column and eluted on AKTA FPLC, followed by desalting using Zeba Spin Desalting Columns (7 K MWCO, 10 mL) according to the manufacturer's protocol. Purified recombinant proteins were analyzed by SDS-PAGE, and concentrated using stirred ultrafiltration cells (Millipore) according to the manufacturer's protocol. The concentration of each protein was determined using 280 nm wavelength on a NanoDrop 2000c (Thermo Scientific).

**Mammalian expression of wild-type DJ-1 and C106A mutant**. The pGW1-Myc-DJ1 plasmid[48] was a kind gift from Mark Cookson (Addgene plasmid # 29347). The Myc-tagged DJ-1 C106A mutation was cloned by inverse PCR using pGW1-Myc-DJ1 as the template and the following primer sequences: 5′-GCCGCCA TCGCCGCAGGTCCTACTGCTCTGTTGGC-3′ and 5′-TATCAGGCCCTTC CGGTTTTCCTGCTCCTTCAGTATC-3′. The Myc-tagged wild-type DJ-1 and DJ-1 C106A mutation were expressed in HEK 293T cells using Lipofectamine 2000 Transfection Reagent (Thermo Fisher Scientific) according to the manufacturer's protocol. DJ-1 overexpression was detected by western blot analysis with anti-DJ-1 and anti-Myc antibodies.

**In vitro glycation assays**. Free histone glycation assays were prepared on ice and contained 20 μM of the designated histone (H2A, H2B, H3.2 or H4) in PBS buffer (pH 7.4) and the corresponding concentrations of MGO (Sigma). Reactions were incubated at 37 °C for 12 h and quenched by flash freezing. Glycated histones were analyzed by SDS-PAGE (without boiling the sample) followed by western blot analysis.

Intact nucleosome gycation assay was similarly prepared using 10 pM NCPs and the corresponding concentration of MGO. The reactions were prepared on ice and incubated at 37 °C for 72 h. The glycated NCPs were immediately analyzed by SDS-PAGE or native page electrophoresis followed by western blot analysis.

Nucleosomal array glycation assay was similarly prepared using 1 μM dodecameric arrays and the corresponding concentration of MGO. The reactions were prepared on ice and incubated at 37 °C for 12 h. The glycated arrays were separated on APAGE, followed by either ethidium bromide staining or western blot analysis.

H3 N-terminal peptide glycation assays contained 0.5 mM peptide in PBS buffer (pH 7.4) and 1.5 mM MGO (corresponding to a 1:3 sites/MGO stoichiometry). The reaction was prepared on ice and incubated at 37 °C for 30 min.

**MNase digestion of nucleosomal arrays**. Micrococcal nuclease (MNase) was purchased from Worthington Biochemical Corporation and the digestion assays were performed according to the manufacturer's protocol with slight modification. Briefly, 1 μL MNase (0.2 U) was added to 2 pmoles of nucleosomal arrays in digestion buffer (50 mM Tris pH 7.9, 5 mM CaCl$_2$). Following, the reaction was quenched with quenching buffer (0.4 M NaCl, 0.2 % (w/v) SDS, 20 mM EDTA) and the DNA purified by Spin Miniprep Columns (Qiagen). The purified DNA was analyzed by gel electrophoresis and ethidium bromide staining.

**MgCl$_2$ precipitation of nucleosomal arrays**. The MgCl$_2$ precipitation of nucleosomal arrays was performed according to the published procedure[28]. Briefly, increasing concentrations of MgCl$_2$ were added to the nucleosomal arrays and the reaction was incubated on ice for 10 min, followed by a 10-minute 17,000 rcf spin at 4 °C. The A260 of the supernatant was measured and used to evaluate the fraction of soluble arrays.

**In vitro deglycation assays**. Purified recombinant wild type and mutant C106A DJ-1 were stored in 50 mM Tris-HCl pH 7.5, 100 mM NaCl, 10 % glycerol (v/v) and 1 mM DTT. In all DJ-1 mediated deglycation assays the recombinant DJ-1 was added to the reaction at a 1:10 stochiometry of enzyme: substrate during or after the glycation reaction.

Full length H3 deglycation assay was performed with DJ-1 (or C106A mutant) added to the reaction two hours after the glycation reaction was initiated and incubated at 37 °C for an additional hour.

NCPs and arrays deglycation assay was performed with DJ-1 (or C106A mutant) added to the glycation reaction and co-incubated at 37 °C for 6 h total. The reactions were analyzed by western blot (full-length H3 and NCPs) and MgCl$_2$ precipitation (nucleosomal arrays).

**Analysis of nucleosomal array architecture by optical tweezers**. A 10 kb plasmid derived from pET-28b (Novagen) and pWM531 was linearized, placing 12 × '601' nucleosome positioning sequences at the center. A short oligonucleotide duplex with dual biotin tags was ligated to each end. Nucleosomal arrays were formed by salt dialysis with human histone octamers heterologously expressed in E. coli. Arrays were pre-treated with 100 mM MGO at 4 °C overnight before experiments. Single-molecule experiments were performed on a commercial C-Trap instrument (LUMICKS). Individual arrays were tethered between a pair of 3.23 μm streptavidin-coated polystyrene beads. The tethers were pulled at a constant speed of 0.1 μm/s in a buffer consisting of 10 mM Tris pH 8.0, 0.1 mM EDTA, 200 mM KCl, 0.5 mM MgCl$_2$, and 0.1% Tween-20.

**Salt extraction of histones from cells**. The extraction of histones from cells was performed according to the previously described high salt extraction method[22]. Briefly, the cell lysis was performed by using the extraction buffer (10 mM HEPES pH 7.9, 10 mM KCl, 1.5 mM MgCl$_2$, 0.34 M sucrose, 10% glycerol, 0.2% NP40, protease and phosphatase inhibitors to 1 × from stock). After spinning down, the pellet was extracted using no-salt buffer (3 mM EDTA, 0.2 mM EGTA). After discarding the supernatant, the final pellet was extracted by using high-salt buffer (50 mM Tris pH 8.0, 2.5 M NaCl, 0.05% NP40) in 4 °C cold room for 1 h. After spinning down, the supernatant containing extracted histones was collected for further analyses.

**Co-Immunoprecipitation**. The glycated and unglycated N-terminal biotinylated H3 peptides (residues 1–18) were used in the Co-IP assays as previously described[49]. The glycated peptides were prepared by a treatment of MGO in a 1:3 (peptide: MGO) stoichiometry at 37 °C for 30 min as described above. Recombinant DJ-1 or total 293T cell lysate (prepared as described before[49]) were incubated with the peptides in 4 °C for 2 h after which the peptide was pulled down by BSA-blocked Streptavidin Magnetic Beads (Thermo Scientific). Next, beads were washed 3 times with 1× PBS buffer (pH 7.4), boiled, separated on SDS-PAGE and analyzed by western blot with anti-DJ-1.

**Breast cancer cell lines and tumor samples**. All cell-culture reagents were obtained from Thermo Fisher Scientific unless otherwise indicated. 293T cells (ATCC) and breast cancer cell lines (Chandarlapaty lab, MSKCC) were cultured at 37 °C with 5% CO$_2$ in DMEM medium supplemented with 10% fetal bovine serum (FBS) (Sigma-Aldrich), 2 mM L-glutamine and 500 units mL$^{-1}$ penicillin and streptomycin. The SKBR3 cell line was a kind gift from Dr. Neal Rosen (MSKCC). Cells were treated with corresponding amounts of MGO described in the main text. Briefly, 1 M and 100 mM MGO stock solutions were dissolved in deionized water and then added to the media accordingly.

All cell lines for tumor xenografts were maintained at 37 °C and 5% CO$_2$ in humidified atmosphere. MCF7 was obtained from DSMZ, and CAMA-1 and BT474 were obtained from ATCC. Cells were grown in DMEM/F12 supplemented

with 10% FBS, 100 μg/mL penicillin, 100 mg/mL streptomycin, and 4 mmol/L-glutamine. All the cell lines were tested negative for Mycoplasma and authenticated by short-tandem repeat (STR) analysis. Six-to-8-week-old nu/nu athymic BALB/c female mice were obtained from Harlan Laboratories, Inc., and maintained in pressurized ventilated caging. All studies were performed in compliance with institutional guidelines under an Institutional Animal Care and Use Committee-approved protocol (MSKCC#12–10–016). MCF7 and CAMA-1 xenograft tumors were established in nude mice by subcutaneously implanting 0.18-mg sustained release 17β-estradiol pellets with a 10 g trocar into one flank followed by injecting 1 × 107 cells suspended 1:1 (volume) with reconstituted basement membrane (Matrigel, Collaborative Research) on the opposite side 3 days afterward.

The clinical samples (MSKCC set) used in this study were obtained from the Biobank of MSKCC. The Patients with breast cancer and either recurrence of disease after receiving adjuvant therapy or WHO-defined progression of metastatic disease on therapy were prospectively enrolled on an IRB approved tissue collection protocol (IRB#06-163). Informed consent was obtained from all patients. All patients underwent biopsy of at least a single site to document progressive disease. Mutational analysis of the metastatic biopsy was performed on fresh frozen specimens. Formalin fixed paraffin embedded (FFPE) blocks of the pretreatment primary tumor was obtained where possible for comparison. The presence of tumor, in both frozen samples and FFPE tissue sections, was confirmed by the study pathologist. Western blot analyses of DJ-1 expression and histone glycation were performed on fresh frozen specimens.

**Cell fractionation**. The cytosolic and nuclear fractions were prepared using NE-PER Nuclear and Cytoplasmic Extraction Reagents (Thermo Scientific) according to the manufacturer's protocol. Histones were extracted from the pellet using high salt extraction protocol as described above[22]. Histone extraction from tumor xenografts and patient samples followed a similar protocol, with slight variation, where tumors homogenized by mild sonication prior to extraction. Purity of fractionation was evaluated using the following antibodies: anti-Actin (cytosol), anti-MEK ½ (nucleoplasm) and anti-H3 (chromatin)[50].

**ATAC-sequencing experiment and computational analysis**. The ATAC-seq samples were generated using 293T cells cultured in the presence or absence of 0.5 mM MGO for 12 h, after which the cells were harvested and diluted to the final concentration of 200 cells/μL. 50,000 cells of each sample were immediately processed following the previously published protocol[51] by the Integrated Genomics Operation of Memorial Sloan Kettering Cancer Center. Libraries were sequenced on an Illumina HiSeq 4000 with 50 bp paired end reads to approximately 50 million reads per sample. Sequencing reads were mapped to the human genome (GRCh38.p12) with BWA-MEM v0.7.15[52]. Mitochondria-mapped and multi-mapping reads were removed using Picard Tools v2.18.4 MarkDuplicates. The sequencing depth of each sample was normalized by randomly subsetting reads. Prior to peak calling, the distribution of insert sizes was compared between the samples (Supplementary Figure 3). Broad read peaks were called using MACS2, excluding regions of low mappability defined by ENCODE Project Consortium's GRCh38 blacklist regions[53,54]. Peak calls with a Benjamini-Hochberg q value < 0.01 were used for downstream analysis. Consensus peaks were determined using an Irreproducible Discovery Rate limit of 0.05[55].

NucleoATAC was used to identify high confidence nucleosome-occupied and nucleosome-free regions from the consensus MACS2 peaks[56]. In order to visualize the read distribution around nucleosome peaks and nucleosome-free regions, deepTools v3.0.2 was used to generate bigwig files and plot the heatmaps in Fig. 4g.

**Cell viability assay**. Untreated and 24-hours DJ-1-transfected 293T cells were cultured in a 96-well plate and treated with the indicated MGO concentrations for 12 h. Following the incubation, cell viability was evaluated using the Cell Counting Kit-8 (CCK-8, Sigma) according to the manufacturer's protocol. The relative cell viabilities were given by detecting the absorbance at 460 nm at each well. Each experiment was performed in triplicate.

**Inhibitor treatment**. The HAT inhibitor curcumin (50 μM)[57], HDAC inhibitor SAHA (5 μM)[58] and MGO scavenger Carnosine (10 mM)[10] were added to the cells' media 2 h prior to adding the corresponding concentrations of MGO. Cells were incubated for an additional 12 h after which they were harvested and histones were extracted and analyzed as described above. Samples were separated on a single SDS-PAGE, transferred to a PVDF membrane and blotted with the indicated antibodies.

**Pulse-chase experiments**. 293T cells were treated with 1 mM MGO for 12 h before the medium was changed to MGO-free DMEM (or 36 h without medium change). Cells were cultured for an additional 12 or 24 h after which they were harvested and cytosolic and histone fractions were prepared as described above. Samples were separated on a single SDS-PAGE, transferred to a PVDF membrane and blotted with the indicated antibodies.

**shRNA knockdown**. The plasmid sh-DJ1 was a kind gift from Hans-Guido Wendel's lab at MSKCC, which was purchased from Sigma and constructed by the previously

reported methods[59]. The shRNA plasmids were transfected with Lipofectamine 2000 Transfection Reagent (Thermo Fisher Scientific) according to the manufacturer's protocol. Briefly, cultured 293T cells were washed with Opti-MEM medium (Invitrogen) and then transfected with the shRNA plasmid using Lipofectamine 2000 Transfection Reagent in Opti-MEM medium without serum. 6 h after transfection, the culture medium was replaced with fresh complete DMEM medium. The cells were subjected to further treatment and/or analysis 24 h after transfection.

**Reporting Summary**. Further information on experimental design is available in the Nature Research Reporting Summary linked to this article.

## Data availability

Raw and processed sequencing data are available from the Gene Expression Omnibus (GEO) database under accession code GSE121252. All the other data are available within the article and its Supplementary Information files or from the corresponding author upon reasonable request. A reporting summary for this Article is available as a Supplementary Information file.

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

## Acknowledgements

We would like to thank Dr. Zachary Brown and Prof. Benjamin A. Garcia for major intellectual contributions. We would also like to thank all members of the David lab for valuable discussions and especially Nicholas A. Prescott for critical reading of the manuscript. A.O. is supported by the National Science Foundation Graduate Research Fellowship Grant Number 2016217612. A.O. and R.L. are supported by the NIH T32 GM115327 Chemical-Biology Interface training grant. N.D.O. is supported by the Tri-Institutional Training Program in Computational Biology and Medicine (via NIH training grant 1T32GM083937). S.C. is supported by funds from the BCRF. This work was supported by the Josie Robertson Foundation, SPORE NIH grant P50-CA192937 and the CCSG core grant P30 CA008748. We acknowledge the use of the Integrated Genomics Operation Core, funded by the NCI Cancer Center Support Grant (CCSG, P30 CA08748), Cycle for Survival, and the Marie-Josée and Henry R. Kravis Center for Molecular Oncology.

## Author contributions

Q.Z., and Y.D. designed the research; Q.Z., N.D.O., R.L., A.O., A.S.A., E.F., H.D., and B.L. performed the experiments; Q.Z., N.D.O., S.C., S.L. and Y.D. analyzed the data; Q.Z., and Y.D. wrote the paper with the other authors' help. Y.D. directed the research.

## Additional information

**Competing interests:** S.C. has received research funds from Daiichi-Sankyo (to institution), Novartis (to institution), Sanofi (to institution), Genentech (to institution) and Eli Lilly (to institution) and ad hoc consulting honoraria from Novartis, Sermonix, Eli Lilly, Context Therapeutics, and Revolution Medicines. All the other authors declare no competing interests.

