## [Peer Review File · Nature Communications]

Reviewers' comments:

Reviewer #1 (Remarks to the Author):

In this interesting paper Yael David et al describe the non-enzymatic modification of histones by the reactive aldehyde methyl Glyoxal (MG). MG is known to be produced in cells through a variety of biochemical pathways, particularly those associated with glucose metabolism. There is growing interest in MG as it is thought to promote protein glycation and recently also DNA glycation, the consequences of these modification are currently unclear. However an enzyme called DJ-1 acts to counteract the effects of MG, since it removes MG adducts on proteins and DNA bases, humans deficient in DJ-1 develop neurodegeneration. Briefly Yael et al show that MG glycates histones in vitro and in vitro (cellular), though high doses of MG probably way outside the physiological range are used mM range. They show that glycated histones are altered in there properties and they show that DJ-1 can remove MG adducted histones. This is a provocative study reporting a potentially important finding, considering how ubiquitous MG is there is very little research in this field and this paper is one of few. On the whole I have little criticism of the experiments performed which are convincing, however for this paper to be a strong candiate for Nature communications this reviewer would like to see more functional evidence.

1. Do Dj-1 deficient cells show epigenetic dysfunction or indeed show any dysfunction that can be ascribed to histone glycation ?
2. What happens when DJ-1 is knocked out in high DJ-1 expressing cell lines, can one then see a massive increase in histone glycation and are these associated with any phenotype.
3. The enzyme Glo-1 removes MG, does Glo-1 knockout cells show increased levels of histone glycation ?

Reviewer #2 (Remarks to the Author):

SUMMARY

This study focusses on the emerging importance of methylglyoxal (MGO) in protein glycation and cell dysfunction. Important advances have been made in recent years and MGO glycation may have role in carcinogenesis and cancer chemotherapy. The authors are inexperienced in glycation research and have made multiple errors of experimental design and choice of methodology throughout. Combined with decisions taken in experimental design to use MGO concentrations of extreme suprphysiological concentration (1000 - 50,000 fold higher than physiologically relevant), this has produced a study that has produced artefactual outcomes or outcomes only relevant to acute MGO intoxication. All of the current studies require repeating with high purity MGO at physiologically relevant concentrations and with reliable and robust established and validated sample processing and analysis techniques for glycation research. I describe critical flaws and remedial action required below.

INTRODUCTION

Comment: There are some incorrect descriptions and misunderstandings on glycation here. Methylglyoxal (MGO) is not a glycolytic intermediate but rather a by-product formed from two glycolytic intermediates. The guanidino group of arginine residues that participates in glycation is not an amino group. The formation of the major advanced glycation endproduct (AGE) quantitatively, MGO-derived hydroimidazolone MG-H1, and other AGEs are formed non-oxidatively. It should be stated that the major AGEs quantitatively, arginine-derived hydroimidazolones, have slow chemical dynamic reversibility; for example, MG-H1 reverses spontaneously on long-lived proteins with a half-life of about 12 days. So it is not correct to generally state that formation of AGEs cannot be reversed.

It should be stated that the early attempt to assay AGEs in histone proteins by fluorescence

(reference 15), now known to be not specific nor quantitative for AGE content, requires re-assessment. Similarly, the study in reference 16 lacks physiological relevance – using D-glucose 500 mM.

RESULTS

IN VITRO GLYCATION OF HISTONES WITH MGO

Comment: The SDS-PAGE analysis is indicating oligomerisation – including of peptide backbone fragmentation products (producing the peptides over a continuous and wide range of molecular masses). This is an artefact of the methodology used.

Firstly, in vitro glycation experiments have been performed with MG concentrations at least ≥ 20 -fold higher than found physiologically and with commercial MGO which is known to contain high levels of crosslinking agent, formaldehyde [1].

Secondly, the samples were also lyophilised without prior removal of MGO by dialysis or diafiltration. Lyophilisation under these conditions forces MGO onto protein by chemical dehydration in the lyophilisation process.

Thirdly, the samples were likely heated denaturation in SDS-PAGE analysis. This has likely produced the peptide backbone fragmentation and formation of further glycation adducts.

Glycation adduct content should be analysed by a robust technique – such as stable isotopic dilution analysis LC-MS/MS after exhaustive enzymatic hydrolysis.

These studies must be repeated with high purity MGO at physiological concentrations (2 – 10 micromolar) and up to no more than 10 –fold above this (100 micromolar), washing protein free of MGO prior to lyophilisation and analysis of AGE adducts by LC-MS/MS. The current data are most likely an artefact of using impure MGO and failing to remove the MGO from the protein before lyophilisation and sample heating. I have performed similar experiments under the appropriate methodology indicated and found no evidence of high molecular mass proteins/polypeptide formation. I have to conclude that the data in Figure S1 and studies of glycated histones produced thereby are artefacts and these studies require repeating with appropriate methodology.

Page 5, Paragraph 1, Figure 1b – Page 6, line 9

Comment: These experiments require repeating – see above. The current SDS-PAGE gel analysis is an artefact of the methodology used.

The comment that the cellular concentrations of MGO are unknown but 0.32 mM has been measured is an ill-informed comment on this research field. The estimate of 0.32 mM MGO has long been known to be an artefact. Robust methods of MGO analysis are available and cellular and tissue concentrations of MGO are 2 – 10 micromolar [2]. The experiments with 0.25-1 mM MGO are therefore using concentrations of 25 – 100 fold higher than the upper limit found physiologically – and with commercial, impure MGO. These studies require repeating with physiologically relevant levels of high purity MGO. The data will not be considered further by this reviewer as the results are an artefact of this current inappropriate experimental design and implementation.

The level and fates of MG-modified proteins in cells round the cell cycle should be assessed by assessment of MG-derived AGEs and proteomics analysis of MG-modified proteins for which reliable, robust and validated techniques have been developed [3, 4]. The authors unfortunately use inappropriate experimental design and implementation.

HISTONE GLYCATION DISRUPTS NUCLEOSOME ASSEMBLY AND STABILITY

Comment:

Paragraph 1. These data are unreliable and the study requires repeating with appropriate experimental design and methodology – see above.

Paragraph 2. Using 2, 30 and 100 mM MGO the authors have used MGO concentrations 1000, 15,000 and 50,000 higher than is physiologically relevant. The authors unfortunately use inappropriate experimental design and implementation. These data are unreliable and the study requires repeating with appropriate experimental design and methodology

HISTONE GLYCATION DISRUPTS CHROMATIN ARCHITECTURE IN VITRO AND IN CELLULO

Paragraph 1. Using 100 mM MGO the authors have used MGO concentrations 50,000 higher than is physiologically relevant. Again, the authors unfortunately use inappropriate experimental design and implementation. These data are unreliable and the study requires repeating with appropriate experimental design and methodology

DJ-1 ACTIVELY DEGLYCATES HISTONES IN VITRO AND IN CELLULO

Comment: It remains uncertain if DJ-1 has a role in protein glycation when physiological levels of glycation adducts are studied. Again, the authors unfortunately use inappropriate experimental design and implementation. These data are unreliable and the study requires repeating with appropriate experimental design and methodology

HISTONES ARE BASALLY GLYCATED IN BREAST CANCER

Comment: AGE adduct should be measured by a robust LC-MS/MS method as the immunoassay method used herein is not specific and unfit for purpose as not AGEs can be detected in control, MGO-untreated cells. Using 0.25, 0.5 or 1 mM MGO, again extreme supraphysiological levels of MGO have been used. Again, the authors unfortunately use inappropriate experimental design and implementation. These data are unreliable and the study requires repeating with appropriate experimental design and methodology.

The way to see an effect of carnosine, of course, is to use high supraphysiological levels of MG and extracellular carnosine – which is done here – to force interaction by experimental design rather than physiologically relevant test. Again, the authors unfortunately use inappropriate experimental design and implementation. These data are unreliable and the study requires repeating with appropriate experimental design and methodology.

DISCUSISON

Comment: This is based on generally an artefactual experimental data set. I have not read this since it will change profoundly when experiments are repeated with appropriate design and methodology.

References

- [1] J Biol Chem 261: 14240-14244
- [2] Nature Protocols 9: 1969-1979
- [3] Biochem Soc Trans 42: 511-517
- [4] Glycoconjugate Journal 33: 553-568

Reviewer #3 (Remarks to the Author):

The paper entitled “Reversible histone glycation drives disease-associated changes in chromatin architecture” by David and co-workers describes and characterizes an exciting new link between cellular metabolism and epigenetic regulation. Specifically, the authors show that histone proteins can undergo non-enzymatic glycation with the reactive glycolytic intermediate methylglyoxal (MGO). They go on to demonstrate that histone glycation has significant impact on other histone PTMs, on the assembly and stability of nucleosomes, and chromatin structure overall. While these observations by themselves already constitute a remarkable advance, the tour-de-force continues to the characterization of the reversal of glycation, mediated enzymatically by the histone deglycase DJ-1.

The paper showcases an impressive array of techniques, spanning from peptide and protein chemistry and nucleosome biochemistry, to cancer biology. Applying such an interdisciplinary

approach, the authors are able to reveal a pathophysiological accumulation of histone glycation in cancer. Surely this work will spark wide interest and opens up many fascinating research directions, to further explore the link between metabolic damage, histone modifications, and epigenetic regulation. All experiments are logical, the results clearly described and the conclusions well-founded. The paper is well written and organized. I therefore fully support the publication of this manuscript in Nature Communications, and I believe it will be of broad interest to its readership.

Before publication, I have a few minor comments the authors should consider:

In general, it was in some places hard to follow what concentrations of MGO were used in the experiments (also since it varied quite a bit from experiment to experiment). Using the very high concentration of MGO (100 mM) is a debatable model for protein glycation and should be explained more carefully in the text.

Is it known which glycation products are recognized by the anti MGO antibody? If so, it would be helpful to know that.

Scheme 1: the carbon side chain in Arg should have one carbon atom less.

Figure 1a: figure caption and text says 12h of incubation, but the supporting information states 72h of incubation, please double check.

Comparing the results in Fig. 1a and Fig. 1b, there is a significant increase in the concentration of MGO used to treat the nucleosomes in Fig. 1b. Why the concentration increased so dramatically? Justifying this choice in the main text would be valuable.

It is stated that the exact concentration of MGO in vivo is not known, but in cancer and in CHO cells it is estimated at around 0.32 mM. If the basal concentration is similar in 293T cells, it would be expected that treatment with 0.5 mM MGO would increase MGO just over 2-fold. However, there is no evidence of any background MGO modification in the (-) lane of Fig. 1c. Is this simply a result of the exposure time, or is there genuinely no background MGO modification present in this cell line? And if so, why is that?

Figure 3: Figure 3a: Could the authors comment on the lower amount of DNA in the gel upon MGO treatment?

The scales of the bar graphs in Figure 3b are different, making a direct comparison of the value ranges harder. This is especially important, given that the averages are within error. The statistical test used to establish significance or at least the criteria for significance should be stated in the figure caption.

The data in 3c were compelling, but Figure S4 would also benefit from being in the main text as it supports the argument well (and helps clarify the different choices of MGO concentrations).

Figure 1, caption, line 6: nucleosomes

Reviewer #1:

In this interesting paper Yael David et al describe the non-enzymatic modification of histones by the reactive aldehyde methyl Glyoxal (MG). MG is known to be produced in cells through a variety of biochemical pathways, particularly those associated with glucose metabolism. There is growing interest in MG as it is thought to promote protein glycation and recently also DNA glycation, the consequences of these modification are currently unclear. However an enzyme called DJ-1 acts to counteract the effects of MG, since it removes MG adducts on proteins and DNA bases, humans deficient in DJ-1 develop neurodegeneration. Briefly Yael et al show that MG glycates histones in vitro and in vitro (cellular), though high doses of MG probably way outside the physiological range are used mM range. They show that glycated histones are altered in there properties and they show that DJ-1 can remove MG adducted histones. This is a provocative study reporting a potentially important finding, considering how ubiquitous MG is there is very little research in this field and this paper is one of few. On the whole I have little criticism of the experiments performed which are convincing, however for this paper to be a strong candidate for Nature communications this reviewer would like to see more functional evidence.

1. Do Dj-1 deficient cells show epigenetic dysfunction or indeed show any dysfunction that can be ascribed to histone glycation?

The reviewer raises an important question. To address this, we added new experiments where we knocked down as well as overexpressed DJ-1 in 293T cells and examined the effect this has on histone PTMs (e.g., acetylation and methylation) (**new Figure S10**). Our results indeed show that DJ-1 deficiency promotes even more significant MGO-induced epigenetic damage that can be rescued by its overexpression, returning PTM levels to their normal state. Moreover, we performed a cell viability assay, which revealed that DJ-1 knockdown results in remarkably decreased survival, particularly in the presence of MGO (**new Figure S9**).

2. What happens when DJ-1 is knocked out in high DJ-1 expressing cell lines, can one then see a massive increase in histone glycation and are these associated with any phenotype.

This is an interesting question that may also be important for understanding the therapeutic potential of DJ-1. To address this, we attempted to generate CRISPR-Cas9 mediated *DJ-1*^{-/-} breast cancer cell lines (which naturally express high levels of DJ-1), however, we could not isolate a pure cell line based on the genotypic screen. This result could indicate that DJ-1 is essential for the survival of these breast cancer cell lines. As an alternative method, transient transfection was utilized to deliver the shDJ-1 plasmid to the highest expressing DJ-1 breast cancer cell line, SKBR3. We found that DJ-1 knockdown significantly decreased cell viability (**new Figure S12a**) and in direct correlation, histones extracted from these cells exhibit an increase in overall glycation (**new Figure S12b**).

3. The enzyme Glo-1 removes MG, does Glo-1 knockout cells show increased levels of histone glycation?

Glo-1 is an important metabolic enzyme that can convert free MGO to lactate, preventing its accumulation and thus reaction with other cellular molecules. Glo-1 can rescue MGO-induced damage in this indirect manner, and is indeed an active line of investigation in the lab. However, for this manuscript, we focused on DJ-1, which is a direct deglycase that can act on glycated histones. Interestingly, during the review process, and in support of our findings, a metabolomics paper was published that illustrates, as the reviewer proposed, that knocking down Glo-1 increases histone glycation (*PNAS*, 2018, 115(37), 9228-

9233). We have included a Glo-1 in the manuscript so this part of the introduction now reads:

“Thus, it is not surprising that various cellular mechanisms, such as Glo-1 and carnosine, have evolved to prevent MGO accumulation (17). Moreover, recent evidence suggests enzymatic reversibility of early glycation intermediates (Scheme 1) although there is no known correction machinery for crosslinked AGEs (10, 11).”

Reviewer #2:

SUMMARY

This study focusses on the emerging importance of methylglyoxal (MGO) in protein glycation and cell dysfunction. Important advances have been made in recent years and MGO glycation may have role in carcinogenesis and cancer chemotherapy. The authors are inexperienced in glycation research and have made multiple errors of experimental design and choice of methodology throughout. Combined with decisions taken in experimental design to use MGO concentrations of extreme supraphysiological concentration (1000 - 50,000 fold higher than physiologically relevant), this has produced a study that has produced artefactual outcomes or outcomes only relevant to acute MGO intoxication. All of the current studies require repeating with high purity MGO at physiologically relevant concentrations and with reliable and robust established and validated sample processing and analysis techniques for glycation research. I describe critical flaws and remedial action required below.

INTRODUCTION

4. *There are some incorrect descriptions and misunderstandings on glycation here. Methylglyoxal (MGO) is not a glycolytic intermediate but rather a by-product formed from two glycolytic intermediates.*

We thank and agree with the reviewer that by-product is a more accurate definition for MGO and changed it throughout the manuscript. The reason we defined MGO as a glycolytic intermediate is because after it is generated by spontaneous dephosphorylation of glyceraldehyde-3-phosphate (GA3P) and dihydroxyacetone phosphate (DHAP), it can be converted to lactate by Glo-1/Glo-2, and lactate itself can be further converted to pyruvic acid and feed into other metabolic pathways.

5. *The guanidino group of arginine residues that participates in glycation is not an amino group.*

According to the currently accepted mechanism, the aldehyde group of MGO first reacts with the amino group of arginine and then the MGO ketone group reacts with the arginine guanidino group to form the five-member ring glycation product (Scheme 1).

6. *The formation of the major advanced glycation endproduct (AGE) quantitatively, MGO-derived hydroimidazolone MG-H1, and other AGEs are formed non-oxidatively, It should be stated that the major AGEs quantitatively, arginine-derived hydroimidazolones, have slow chemical dynamic reversibility; for example, MG-H1 reverses spontaneously on long-lived proteins with a half-life of about 12 days. So it is not correct to generally state that formation of AGEs cannot be reversed.*

We apologize for the confusing description of AGEs in the introduction. By definition, advanced glycation endproducts (AGEs) are crosslinked end products rich in aromatic rings or other stable linear structures (e.g., carboxymethyl and carboxyethyl lysine) and cannot undergo any further conversions or be revised by any known enzymes (*Angew Chem. Int. Ed. Engl.*, 2014, 53(39), 10316-10329). Thus, MG-H1 is

by definition not an AGE but a glycation intermediate. We revised this section in the text to now read:

“The initial glycation adduct can further oxidize and rearrange to form a series of stable products, which can undergo additional chemical transformations including the ability to form crosslinks, yielding species generally referred to as advanced glycation end-products (AGEs) (1, 3).”

7. It should be stated that the early attempt to assay AGEs in histone proteins by fluorescence (reference 15), now known to be not specific nor quantitative for AGE content, requires re-assessment. Similarly, the study in reference 16 lacks physiological relevance – using D-glucose 500 mM.

We agree with the reviewer that very little is known about histone glycation and its effect on chromatin structure and function, which makes our manuscript a pioneering study. However, there have been a few attempts to investigate this interesting metabolism-epigenetics link, and as sparse as they are, we thought it is our obligation to mention them. Reference 16 is an *in vitro* study of histone glycation using glucose at high concentration. While we agree this concentration is significantly above the physiological concentration, it is the authors' way of pushing the reaction as often done in biochemical studies (enzymatic and non-enzymatic). For this type of *in vitro* assay, the absolute amount of each component is often amplified while maintaining similar stoichiometry. In the *in vitro* histone glycation assay mentioned in Ref 16, the authors used 500 µg histones as substrates with the final concentration of 5 µg/µL (approximately 0.25 mM), which is potentially higher than the physiological concentrations of histones in cells. There are about 50 Lysine/Arginine residues in H1, so the approximate ratio of glucose and the reactive sites of histones is 40: 1 which is reasonable based on the glycation reactivity of glucose. Regarding reference 15, we did not perform fluorescence analysis to quantify histone glycation in our analysis.

RESULTS

IN VITRO GLYCATION OF HISTONES WITH MGO

8. The SDS-PAGE analysis is indicating oligomerisation – including of peptide backbone fragmentation products (producing the peptides over a continuous and wide range of molecular masses). This is an artefact of the methodology used.

As far as we know, MGO glycation cannot cause the cleavage or fragmentation of target proteins. We do not have a western blot analysis of peptide glycation (or protein glycation post digestion) in our manuscript, so there are no peptide fragmentation adducts in our assays. The laddering we observe in protein and nucleosome glycation is due to a variety of glycation adducts including crosslinking as these species are recognized by both anti MGO and anti H3 (also see the LC-MS analysis in **Figure R3**, below).

9. In vitro glycation experiments have been performed with MG concentrations at least ≥ 20 -fold higher than found physiologically and with commercial MGO which is known to contain high levels of crosslinking agent, formaldehyde [1].

The MGO solution we used throughout the manuscript was purchased from Sigma-Aldrich (catalog number 67028), is highly pure, and was used in more than 40 peer-reviewed papers. The reference the reviewer mentions (*J. Biol. Chem.*, 1986, 261(30), 14240-14244) did not designate the source or purity of the MGO used in their study, however, we agree this could have been a poor quality MGO as the experiments were performed more than 30 years ago. Although the purity of our MGO is high and as far as we know, MGO cannot be spontaneously convert to formaldehyde, we performed a new experiment where we treated

histone H3 with either MGO or formaldehyde under the same conditions followed by western blot analysis to address the reviewer's concern (**Figure R1**, below). Our results clearly indicate that the anti-MGO antibody we used in our manuscript is unable to recognize the formaldehyde-dependent crosslinking of proteins, **meaning that our observed crosslinkings are MGO-dependent.**

Figure R1. Western bolt analysis of 0.5 mM formaldehyde or MGO-treated H3. All the experimental procedures were performed as described for Figure S1.

We recognize that *in vitro* glycation reactions are different from ones occurring in cells for a variety of reasons including: (1) The concentration of histones in the *in vitro* reactions is higher than that in cells, additionally, in the nucleus histones are protected through interactions and higher-order structures, leaving only a defined subset available for chemical reactions; (2) In cells, histones have sub-localizations which varies their local concentration (e.g euchromatin and heterochromatin), while in the tube histones are evenly distributed; (3) The reaction time allowed for *in vitro* histone glycation is much shorter than that happening in cells (which could reach years). However, in order to dissect the precise biophysical effect glycation has on chromatin, we had to turn to a biochemically defined setup.

As mentioned in our answer to Reviewer 2 Point 4, *in vitro* biochemical assays are almost exclusively performed in absolute concentrations higher than their physiological ones (that includes enzymes, substrates, co-factors, *etc.*). In this case, the chemical, non-enzymatic reaction is primarily affected by the concentration of the reactants and time of reaction (assuming all other conditions are similar: temperature, pH, *etc.*). Since glycation events occur over a long period of time (weeks, months and even years), which is a time scale relevant to the half-life of histones, we used concentrations higher than predicted to be physiological in order to expedite these reactions. Our assumption indeed seems to be relevant as nucleosomal arrays incubated with high concentration of MGO for a short time showed a similar compaction perturbation to arrays treated with a short concentration for a longer time (**Figure 3c**). While performed on isolated proteins/protein complexes, the high-resolution mechanistic information we were able to extract from these *in vitro* analyses would be impossible to be obtained in a cellular setup. We agree that standing alone these results might not seem physiologically relevant, but when taken as part of the composite that includes analysis of chromatin in culture (basal glycation in breast cancer cell lines), animal models (mouse xenografts) and human samples (breast cancer patients' tumors), these results are convincing. It is important to note that while high, these are universally acceptable concentrations and ratios to be using for *in vitro* MGO glycation analysis (*J. Biol.*

Chem., 2015, 290(3), 1885-1897; *Science*, 2017, 357(6347), 208-211; *Nature*, 2018, doi: 10.1038/s41586-018-0622-0). To clarify this, we have added an explanation in the text that now reads:

“To test the reactivity of MGO and chromatin components *in vitro* we applied a range of MGO concentrations corresponding to different sites:MGO ratios. These serve to both expedite the glycation reaction (that generates adducts accumulating over months to years *in vivo*) as well as mimic a variety of intermediates generated under different exposure conditions and times.”

10. The samples were also lyophilised without prior removal of MGO by dialysis or diafiltration. Lyophilisation under these conditions forces MGO onto protein by chemical dehydration in the lyophilisation process. Thirdly, the samples were likely heated denaturation in SDS-PAGE analysis. This has likely produced the peptide backbone fragmentation and formation of further glycation adducts.

We want to thank the reviewer for the elaborated technical comments and we apologize for not being clearer with describing our experimental procedure. To clarify, as detailed in our material and methods section in the supplementary material, **we did not lyophilize or boil our histone samples prior to analysis**, being aware of the reactivity of our components. Boiling is also not necessary in our case since the histones we used in our *in vitro* assays do not contain any cysteine residues to reduce.

For DJ-1 deglycation assays, to avoid glycation of DJ-1, excess MGO was removed by centrifugal filter units (or desalting columns) before the addition of the enzyme as previously described (*J. Biol. Chem.*, 2015, 290(3):1885-1897).

11. Glycation adduct content should be analysed by a robust technique – such as stable isotopic dilution analysis LC-MS/MS after exhaustive enzymatic hydrolysis.

We agree with the reviewer, that mass spectrometry (ms) analysis, can provide important high-resolution analysis of histone glycation. Indeed, during the review process, a metabolomics paper was published that reported the mass spectral analysis of histone glycation (*PNAS*, 2018, 115(37), 9228-9233) that supports our findings. Moreover, the MGO concentrations used in this paper were identical to those we used in our analyses.

The reason we were reluctant to use ms in our analyses was the following key disadvantages in applying it to histone glycation: (1) Sample processing (particularly for histone proteins that are highly rich in lysines and arginines) can tamper the results as it relies on anhydride and ammonia treatment (2) Standard ms is not suitable for analyzing crosslinked protein products due to the limitation of ms fragment database establishment (3) After glycation most of the lysines and arginines are unavailable for trypsin digestion as they are blocked by MGO which generates peptides too long for this analysis; (4) Glycated peptides are less charged and thus do not easily get ionized or “fly” on the instrument (5) It is extremely challenging to identify glycated peptides as the products of MGO-treated proteins are very diverse and need to be specifically targeted (see **Figure R2**, below). In fact, when we performed low-resolution ms analysis of MGO-treated H3 (**Figure R3**, below), our results indicated not only the high reactivity of MGO and histones but also how challenging it is to characterize the products by ms. While western blot has its disadvantages, primarily lower sensitivity than ms, we believe in this case the advantages prevail: (1) It is efficient and reliable; (2) The antibodies can be customized to recognize different glycation products in different stages (*Bioorg. Med. Chem. Lett.* 2015, 25(21), 4881-4886) (3) the promiscuous antibody recognizes many sites and forms of MGO-glycation (details of the antibody can be found in our supplementary information).

Arg

Lys

Figure R2. Structures of MGO-mediated protein glycation products, not including crosslinked products and glycation products on other reactive residues (Cys, His, Pro and *etc.*).

Figure R3. LC-MS analysis of WT recombinant (a) and MGO-treated H3 (b). All the experimental procedures were performed as described for Figure S1.

12. *These studies must be repeated with high purity MGO at physiological concentrations (2 – 10 micromolar) and up to no more than 10 –fold above this (100 micromolar), washing protein free of MGO prior to lyophilisation and analysis of AGE adducts by LC-MS/MS.*

Please see comments 9, 10 and 11.

13. *The current data are most likely an artefact of using impure MGO and failing too remove the MGO from the protein before lyophilisation and sample heating.*

Please see comment 10.

14. *I have performed similar experiments under the appropriate methodology indicated and found no evidence of high molecular mass proteins/polypeptide formation.*

Please see comment 11 and Figure R2.

15. *I have to conclude that the data in Figure S1 and studies of glycated histones produced thereby are artefacts and these studies require repeating with appropriate methodology. Page 5, Paragraph 1, Figure 1b – Page 6, line 9. These experiments require repeating – see above. The current SDS-PAGE gel analysis is an artefact of the methodology used.*

Please see comments 8, 9, 10 and 11.

16. *The comment that the cellular concentrations of MGO are unknown but 0.32 mM has bene measured is an ill-informed comment on this research field. The estimate of 0.32 mM MGO has long been known to be an artefact. Robust methods of MGO analysis are available and cellular and tissue concentrations of MGO are 2 – 10 micromolar [2]. The experiments with 0.25-1 mM MGO are therefore using*

concentrations of 25 – 100 fold higher than the upper limit found physiologically – and with commercial, impure MGO. These studies require repeating with physiologically relevant levels of high purity MGO. The data will not be considered further by this reviewer as the results are an artefact of this current inappropriate experimental design and implementation.

In cellulo glycation experiments

First, it is important to note that **the MGO concentrations we used in our experiments, are all consistent with previous published reports** (*Science*, 2017, 357(6347), 208-211; *J. Biol. Chem.*, 2015, 290(3):1885-1897; *PNAS*, 2018, 115(37), 9228-9233; *eLife*, 2016, 5, pii: e19375; *Nature*, 2018, doi: 10.1038/s41586-018-0622-0). However, we recognize that the physiological concentration of MGO inside cells, as many other metabolites, remains a source of controversy. The reference mentioned by the reviewer for low μM concentration of MGO is in fact measured in cell lysate, culture media or blood serum, which are very different from intracellular levels (*eLife*, 2016, 5, pii: e19375). Similarly, information from solid biological samples (tissues or cultured cells) is difficult to interpret as it must be pelleted and lysed prior to LC-MS/MS analysis, so final units are reported in $\mu\text{mol/g}$ and not μM . In addition, MGO is produced in the cytoplasm and diffuses to the nucleus. The local concentration of MGO in nuclei could thus be much different and potentially be measured *in situ* by a “turn-on” fluorescent MGO sensor (*J. Am. Chem. Soc.*, 2013, 135(33), 12429-12433), which, although very intriguing, is beyond the scope of our manuscript.

While it is difficult to estimate the exact concentration of MGO in the nucleus, we agree with the reviewer that **we are using the highest reported physiological concentration** (*PNAS*, 1998, 95(10), 5533-5538) that was used in many other recently published papers (*PNAS*, 2018, 115(37), 9228-9233; *eLife*, 2016, 5, pii: e19375; *Nature*, 2018, doi: 10.1038/s41586-018-0622-0). This concentration is used in order to shorten the experimental time and mimic a long-term exposure to lower MGO concentrations. Our analysis of untreated **human tumor samples** and **mouse xenografts**, which is in complete agreement with our *in vitro* and *in cellulo* experiments serve as the most appealing and direct evidence that histones undergo MGO glycation *in vivo*.

17. The level and fates of MG-modified proteins in cells round the cell cycle should be assessed by assessment of MG-derived AGEs and proteomics analysis of MG-modified proteins for which reliable, robust and validated techniques have been developed [3, 4]. The authors unfortunately use inappropriate experimental design and implementation.

Please see comments 8, 9, 10, 11 and 16.

HISTONE GLYCATION DISRUPTS NUCLEOSOME ASSEMBLY AND STABILITY

18. Paragraph 1. These data are unreliable and the study requires repeating with appropriate experimental design and methodology – see above.

Please see comments 8, 9, 10 and 11.

19. Paragraph 2. Using 2, 30 and 100 mM MGO the authors have used MGO concentrations 1000, 15,000 and 50,000 higher than is physiologically relevant. The authors unfortunately use inappropriate experimental design and implementation. These data are unreliable and the study requires repeating with appropriate experimental design and methodology

Please see comments 8, 9, 10 and 11.

HISTONE GLYCATION DISRUPTS CHROMATIN ARCHITECTURE IN VITRO AND IN CELLULO

20. Paragraph 1. Using 100 mM MGO the authors have used MGO concentrations 50,000 higher than is physiologically relevant. Again, the authors unfortunately use inappropriate experimental design and implementation. These data are unreliable and the study requires repeating with appropriate experimental design and methodology

Please see comments 8, 9, 10 and 11.

DJ-1 ACTIVELY DEGLYCATES HISTONES IN VITRO AND IN CELLULO

21. It remains uncertain is DJ-1 has a role in protein glycation when physiological levels of glycation adducts are studied. Again, the authors unfortunately use inappropriate experimental design and implementation. These data are unreliable and the study requires repeating with appropriate experimental design and methodology

Please see comments 8, 9, 10 and 11.

HISTONES ARE BASALLY GLYCATED IN BREAST CANCER

22. AGE adduct should be measured by a robust LC-MS/MS method as the immunoassay method used herein is not specific and unfit for purpose as not AGEs can be detected in control, MGO-untreated cells. Using 0.25, 0.5 or 1 mM MGO, again extreme supraphysiological levels of MGO have been used. Again, the authors unfortunately use inappropriate experimental design and implementation. These data are unreliable and the study requires repeating with appropriate experimental design and methodology.

Please see comments 8, 9, 10, 11 and 16.

23. The way to see an effect of carnosine, of course, is to use high supraphysiological levels of MG and extracellular carnosine – which is done here – to force interaction by experimental design rather than physiologically relevant test. Again, the authors unfortunately use inappropriate experimental design and implementation. These data are unreliable and the study requires repeating with appropriate experimental design and methodology.

We used carnosine in our assay to illustrate that the effect we observe on histones is MGO-dependent. An important observation we made by using Carnosine on basally glycated breast cancer cell line (SKBR3) is that pre-treating the cells with carnosine abolished the basal histone glycation (**Figure 5b lane 1**) in addition to the sensitivity to MGO treatment. Also, please see comments 8, 9, 10, 11 and 16.

DISCUSSION

24. This is based on generally an artefactual experimental data set. I have not read this since it will change profoundly when experiments are repeated with appropriate design and methodology.

Our discussion explains our experimental design as well as analyzes our results in light of other publications in the field, which address the reviewer's comments.

Reviewer #3:

The paper entitled “Reversible histone glycation drives disease-associated changes in chromatin architecture” by David and co-workers describes and characterizes an exciting new link between cellular metabolism and epigenetic regulation. Specifically, the authors show that histone proteins can undergo non-enzymatic glycation with the reactive glycolytic intermediate methylglyoxal (MGO). They go on to demonstrate that histone glycation has significant impact on other histone PTMs, on the assembly and stability of nucleosomes, and chromatin structure overall. While these observations by themselves already constitute a remarkable advance, the tour-de-force continues to the characterization of the reversal of glycation, mediated enzymatically by the histone deglycase DJ-1.

The paper showcases an impressive array of techniques, spanning from peptide and protein chemistry and nucleosome biochemistry, to cancer biology. Applying such an interdisciplinary approach, the authors are able to reveal a pathophysiological accumulation of histone glycation in cancer. Surely this work will spark wide interest and opens up many fascinating research directions, to further explore the link between metabolic damage, histone modifications, and epigenetic regulation. All experiments are logical, the results clearly described and the conclusions well-founded. The paper is well written and organized. I therefore fully support the publication of this manuscript in Nature Communications, and I believe it will be of broad interest to its readership.

Before publication, I have a few minor comments the authors should consider:

25. *In general, it was in some places hard to follow what concentrations of MGO were used in the experiments (also since it varied quite a bit from experiment to experiment). Using the very high concentration of MGO (100 mM) is a debatable model for protein glycation and should be explained more carefully in the text.*

We apologize for this lack of clarity and thank the reviewer for pointing it out. In the revised manuscript, we added a table detailing the concentrations and ratios of sites:MGO as well as reaction time for the described experiments (**new Table S3**).

We agree with the reviewer that 100 mM MGO concentration is very high and does not model endogenous protein glycation. However, using an *in vitro* setup with a range of MGO concentrations allowed us to access a variety of glycation intermediates and dissect the effect these non-enzymatic adducts have on chromatin at a high resolution, such as biophysical changes in chromatin fiber compaction (see Reviewer 2, comment 9, *In vitro* glycation experiments). We optimized our *in vitro* analyses by utilizing a series of MGO concentrations (for which the corresponding sites:MGO ratios ranged from 1:0.25 to 1:400), in order to model a broad selection of exposure times and MGO concentrations associated with different metabolic states (normal to long-term stress). For example, exposing nucleosomal arrays to either high or low concentration of MGO induced two distinct array compaction states. Importantly, sampling time points of arrays exposed to an even lower MGO concentration revealed the transition between these two states (**Figure 3c**). It is noteworthy that for these reasons, high mM MGO concentrations are commonly used in *in vitro* studies (*J. Biol. Chem.*, 2015, 290(3), 1885-1897; *Science*, 2017, 357(6347), 208-211; *Nature*, 2018, doi: 10.1038/s41586-018-0622-0). We added this explanation to the main text so it now reads:

“To test the reactivity of MGO and chromatin components *in vitro* we applied a range of MGO concentrations corresponding to different sites:MGO ratios. These serve to both expedite the glycation reaction (that generates adducts accumulating over months to years *in vivo*) as well as mimic a variety of intermediates generated under different exposure conditions and times.”

26. Is it known which glycation products are recognized by the anti MGO antibody? If so, it would be helpful to know that.

There are several commercial antibodies that recognize a variety of glycation products at different stages (*Bioorg. Med. Chem. Lett.* 2015, 25(21), 4881-4886). In this study, we chose the most promiscuous antibody with no sequence or MGO-adduct structure specificity as it was generated from serum of animals injected with MGO-treated Ovalbumin, which is very lysine and arginine rich (details of the antibody information are provided in our supplementary information).

27. Scheme 1: the carbon side chain in Arg should have one carbon atom less.

We thank the reviewer for the careful read of our manuscript and for finding this error. We made the correction in the revised manuscript (revised **Scheme 1**).

28. Figure 1a: figure caption and text says 12h of incubation, but the supporting information states 72h of incubation, please double check.

We apologize for the confusion. We used a 12-hour incubation for glycation assays with **free histones (Figure 1a)**, while 72-hour incubation was used for glycation assays with **NCPs (Figure 1b)**. We corrected this discrepancy and updated it in the **new Table S3**.

29. Comparing the results in Fig. 1a and Fig. 1b, there is a significant increase in the concentration of MGO used to treat the nucleosomes in Fig. 1b. Why the concentration increased so dramatically? Justifying this choice in the main text would be valuable.

Free histones are highly amenable to glycation as all the active side chains are exposed (histones only fold in the presence of all 4 core histones that form the symmetric octamer). Thus, a short incubation and low MGO concentration suffice to see intense glycation. In fact, leaving those for longer or at higher MGO concentrations results in aggregation that accumulated at the top of the well due to rearranged crosslinking. Nucleosomes, on the other hand, are composed of a histone octamer (where the histones are protected) and 147 bp of DNA that both protects the histones and can consume some the MGO (as it undergoes glycation itself). To overcome this and see an MGO-concentration dependent adducts we expanded our concentrations up to 30 mM (which is an extreme over saturation, corresponding to 1:120 ratio of sites:MGO) and a 72 hours incubation. However, we could already observe H3 glycation at the lowest MGO concentration, 0.5 mM, which corresponds to 1:2 ratio. In the revised manuscript, we added this explanation that now reads:

“Since many of the sites of modification are hindered in NCPs compared to free histones, either by histone-histone or histone-DNA interaction, we predicted the glycation reaction will be slower. Reconstituted NCPs were thus incubated with higher ratios of MGO for 72 hours, after which they were separated by SDS-PAGE and analyzed by western blot with anti MGO.”

30. It is stated that the exact concentration of MGO in vivo is not known, but in cancer and in CHO cells it is estimated at around 0.32 mM. If the basal concentration is similar in 293T cells, it would be expected that treatment with 0.5 mM MGO would increase MGO just over 2-fold. However, there is no evidence of any background MGO modification in the (-) lane of Fig. 1c. Is this simply a result of the exposure time, or is there genuinely no background MGO modification present in this cell line? And if so, why is that?

We thank the reviewer for bringing up this important point. The intracellular concentration of MGO is cell type- and metabolic state-dependent. It has been reported that MGO concentrations are very high in colon, prostate, lung, melanoma, and breast cancer cell lines (*eLife*, 2016, 5, pii: e19375) due to enhanced Warburg effect. However, in 293T or HeLa cells, the MGO concentration is in fact much lower (*PNAS*, 2018, 115(37), 9228-9233), and so we did not observe any basal histone glycation in 293T cells grown under normal culture conditions.

31. *Figure 3: Figure 3a: Could the authors comment on the lower amount of DNA in the gel upon MGO treatment?*

This is an interesting observation. Since we use a DNA intercalating agent for staining the arrays, we hypothesized that the reason for a lower DNA signal in glycated arrays is that the glycation adducts perturb the intercalation and/or that the glycated arrays are more rigid. In fact, we observed this phenomenon on a mononucleosomal level, which suggest the former (**Figure 2c**).

32. *The scales of the bar graphs in Figure 3b are different, making a direct comparison of the value ranges harder. This is especially important, given that the averages are within error. The statistical test used to establish significance or at least the criteria for significance should be stated in the figure caption.*

We apologize for this lack of consistency. In the revised manuscript we corrected the axes as well as indicated the statistical test in the figure legend.

33. *The data in 3c were compelling, but Figure S4 would also benefit from being in the main text as it supports the argument well (and helps clarify the different choices of MGO concentrations).*

We thank the reviewer for the suggestion. In the revised manuscript, we moved **Figure S4** to **Figure 3c** and adjusted the text accordingly to now read:

“To verify that the higher concentration of MGO mimics a longer exposure we treated arrays with an even lower MGO concentration and tested their compaction state every 6 hours. Indeed, initial time points indicated array decompaction, while later time points exhibited higher compaction, presumably due to rearranging and crosslinking (**Figure 3c**).”

34. *Figure 1, caption, line 6: nucleosomes*

We corrected this typo in the manuscript.

Yael David, Ph.D.

Assistant Member
Chemical Biology Program
Memorial Sloan Kettering Cancer Center
1275 York Ave., Box 428, New York, NY 10065
T 646.888.2127 · davidshy@mskcc.org

Reviewers' comments:

Reviewer #1 (Remarks to the Author):

The authors have adequately addressed my concerns

Reviewer #2 (Remarks to the Author):

SUMMARY

The authors have responded inappropriately to criticism throughout. Their study remains of extremely poor design with use of impure commercial methylglyoxal. The authors mention a recent paper by other investigators robustly quantifying methylglyoxal glycation adducts in vitro and in vivo. The finding was that histone proteins have low, ca. 1% modification by major methylglyoxal-derived glycation adduct MG-H1; crosslinks are much lower than this, usually <0.01%. The current study claiming high levels of methylglyoxal crosslinking of histone proteins is therefore lacking physiological relevance because of inappropriate experimental design and implementation. It is presented as otherwise throughout by the authors which is highly misleading.

BRIEF DETAILED RESPONSES

RESPONSE TO REVIEWER'S COMMENTS

Reviewer 2, point 5 "The guanidino group of arginine residues that participates in glycation is not an amino group."

AUTHORS' RESPONSE: According to the currently accepted mechanism, the aldehyde group of MGO first reacts with the amino group of arginine and then the MGO ketone group reacts with the arginine guanidino group to form the five member ring glycation product (Scheme 1).

FURTHER COMMENT FROM REVIEWER: Unfortunately the authors do not understand or appreciate delocalisation of charge across the two terminal nitrogen atoms and linking carbon in a guanidino group. An amino group does not have this characteristic and hence the guanidino group cannot be described as an amino group. The response is incorrect and an inappropriate response to criticism.

Reviewer 2, point 6. The formation of the major advanced glycation endproduct (AGE) quantitatively, MGO-derived hydroimidazolone MG-H1, and other AGEs are formed non-oxidatively. It should be stated that the major AGEs quantitatively, arginine-derived hydroimidazolones, have slow chemical dynamic reversibility; for example, MG-H1 reverses spontaneously on long-lived proteins with a half-life of about 12 days. So it is not correct to generally state that formation of AGEs cannot be reversed.

AUTHORS' RESPONSE: We apologize for the confusing description of AGEs in the introduction. By definition, advanced glycation endproducts (AGEs) are crosslinked end products rich in aromatic rings or other stable linear structures (e.g., carboxymethyl and carboxyethyl lysine) and cannot undergo any further conversions or be revised by any known enzymes (Angew Chem. Int. Ed. Engl., 2014, 53(39), 10316-10329). Thus, MG-H1 is by definition not an AGE but a glycation intermediate. etc

FURTHER COMMENT FROM REVIEWER: The definition of advanced glycation endproducts (AGEs) as crosslinked end products rich in aromatic rings or other stable linear structures (e.g., carboxymethyl and carboxyethyl lysine) and cannot undergo any further conversions or be revised by any known enzymes is NOT one supported by the article mentioned above by the authors (Angew Chem. Int. Ed. Engl., 2014, 53(39), 10316-10329) NOR generally accepted in the glycation field and beyond. Indeed, in this very article the authors, Hellwig and Henle, describe MG-H1 as an AGE. The authors have presumably read the article and are therefore knowingly providing false evidence with intent to mislead the Editor and reviewers. Clearly, this is an

incorrect and an inappropriate response to criticism.

Reviewer 2, point 7. It should be stated that the early attempt to assay AGEs in histone proteins by fluorescence (reference 15), now known to be not specific nor quantitative for AGE content, requires re-assessment. Similarly, the study in reference 16 lacks physiological relevance – using D-glucose 500 mM.

AUTHORS' RESPONSE: We agree with the reviewer that very little is known about histone glycation and its effect on chromatin structure and function...etc

FURTHER COMMENT FROM THE REVIEWER: The authors did not make the requested comments on the old article cited that used an insecure analytical method for AGE measurement and inappropriate experimental conditions. This an inappropriate response to the criticism.

Reviewer 2, point 8. The SDS-PAGE analysis is indicating oligomerisation – including of peptide backbone fragmentation products (producing the peptides over a continuous and wide range of molecular masses). This is an artefact of the methodology used.

AUTHORS' RESPONSE: As far as we know, MGO glycation cannot cause the cleavage or fragmentation of target proteins. We do not have a western blot analysis of peptide glycation (or protein glycation post digestion) in our manuscript, so there are no peptide fragmentation adducts in our assays. The laddering we observe in protein and nucleosome glycation is due to a variety of glycation adducts including crosslinking as these species are recognized by both anti MGO and anti H3 (also see the LC-MS analysis in Figure R3, below).

FURTHER COMMENT FROM THE REVIEWER: In Figure 1 the authors show bands with a continuous mass range of polypeptides. This does not reflect oligomers which would rather show discontinuous multi-banded ladders bands with molecular masses of multiples of the molecular mass of the oligomer. The continuous band reflects inappropriate sample processing – heating to high temperatures during pre-analytic processing, in indicated in the initial review. Rather than correct this technical flaw, the authors have sort to mislead. The small molecular mass increments of glycation adducts (<100 Da) do not produce discernible mass shift on this mass resolving range of polyacrylamide gels. This an inappropriate response to the criticism.

Reviewer 2, point 9. In vitro glycation experiments have been performed with MG concentrations at least ≥ 20 -fold higher than found physiologically and with commercial MGO which is known to contain high levels of crosslinking agent, formaldehyde [1]. The AUTHORS' RESPONSE: MGO solution we used throughout the manuscript was purchased from Sigma-Aldrich (catalog number 67028), is highly pure, and was used in more than 40 peer-reviewed papers. The reference the reviewer mentions (J. Biol. Chem., 1986, 261(30), 14240-14244) did not designate the source or purity of the MGO used in their study, however, we agree this could have been a poor quality MGO as the experiments were performed more than 30 years ago etc."

FURTHER COMMENT FROM THE REVIEWER: The authors provide a counter claim that the methylglyoxal that they used was highly pure because it was purchased from Sigma Aldrich or "as far as we know" – rather than providing analytical data. This defence of obliviousness may be allowed in a first submission but to return with a revision without addressing the concern is unacceptable. In a recent Nature Protocol method on the measurement of methylglyoxal is was accepted by leading experts in peer review that methylglyoxal as supplied by Sigma is highly impure and a method was given for high purity methylglyoxal preparation (Nature Protocols 9: 1969-1979, 2014). The authors are being disingenuous and seeking to defend an indefensible position. If they believe the methylglyoxal used herein then they must provide analytical data in support of this. There is no substitute here for repeating the experiments with high purity methylglyoxal.

The authors seek to defence the use of supraphysiological concentrations of methylglyoxal because "glycation reactions occur over a long period of time". This fails to observe fundamental concepts in pharmacology of the concentration-response relationship. If the authors are claiming physiological relevance of their research then physiologically relevant concentrations of methylglyoxal must be used. The current studies are only relevant to acute intoxication with very high dose of exogenous methylglyoxal (impure).

Reviewer 2, Point 11. Glycation adduct content should be analysed by a robust technique – such as stable isotopic dilution analysis LC-MS/MS after exhaustive enzymatic hydrolysis.

We agree with the reviewer...etc

FURTHER COMMENT FROM THE REVIEWER: The authors agreed with the reviewer and then do no further work to provide robust analytical data. This an inappropriate response to the criticism.

Reviewer 12 - 22. Relating to the use of extremely high, supraphysiological concentrations of methylglyoxal throughout whilst climaxing physiological significance. The authors have not performed requested experiments with physiologically methylglyoxal concentrations nor assayed methylglyoxal adduct content of their modified proteins. They cite independent work in a recent paper with robust estimation of MG adduct content in histone proteins (PNAS, 2018, 115(37), 9228-9233

FURTHER COMMENT FROM THE REVIEWER: The attempts to justify use of high supraphysiological concentrations of methylglyoxal by citing previous work that has similar poor design and claiming that methylglyoxal (molecular mass 72 Da) may accumulate in the nucleus and not pass through the nuclear membrane (molecular mass cut-off of at least 30,000 Da) seem, frankly, a desperate attempt to publish than to do research of high quality and impact. The arguments do not stand up to scientific scrutiny.

In the paper cited by the authors above on MG-H1 adducts in histone proteins (PNAS, 2018, 115(37), 9228-9233), the glycation adduct content of histone protein in vitro and in vivo was ca. 0.5 pmol per nmol leu, equivalent to ca. 1% histone modification i.e. 1 in every 100 histone proteins bearing one MG-H1 modification. MG-derived crosslinks are much lower than this - <1% of total methylglyoxal modifications (one modification in every 10,000 histone molecules). The highly oligomerised, methylglyoxal-modified proteins produced herein lack physiological relevance. To claim otherwise to misleading and deceiving.

Reviewer 2, point 23. The way to see an effect of carnosine, of course, is to use high supraphysiological levels of MG and extracellular carnosine – which is done here – to force interaction by experimental design rather than physiologically relevant test. Again, the authors unfortunately use inappropriate experimental design and implementation. These data are unreliable and the study requires repeating with appropriate experimental design and methodology.

AUTHORS' RESPONSE: We used carnosine in our assay to illustrate that the effect we observe on histones is MGO-dependent. etc

FURTHER COMMENT FROM THE REVIEWER: Carnosine is not a specific scavenger of methylglyoxal and participates in many other types of reactions. Hence, the design is flawed. The original and now additional critique stand. These data are unreliable and the study requires repeating with appropriate experimental design and methodology – using a specific scavenger of methylglyoxal. This is an inappropriate response to the critique. The is an inappropriate and incorrect response to criticism.

DISCUSSION

Reviewer 2, point 24. This is based on generally an artefactual experimental data set. I have not read this since it will change profoundly when experiments are repeated with appropriate design and methodology.

AUTHORS' RESPONSE: Our discussion explains our experimental design as well as analyzes our results in light of other publications in the field, which address the reviewer's comments.

FURTHER COMMENT FROM THE REVIEWER: The authors have not addressed the flaws in this work and have sought to defend and describe severely flawed research rather than repeat experimental work with appropriate methodology and design. In other responses they have sought to mislead. The original critique therefore stands.

Reviewer #3 (Remarks to the Author):

The authors have submitted a well revised version of their manuscript. They have answered all of my questions, and have provided additional explanations/clarifications where required. I am happy to accept this revised version as is and have no further remarks. Congratulations on a great story!

Reviewer #1 (Remarks to the Author):

The authors have adequately addressed my concerns.

We are happy to see that Reviewer 1 is satisfied with our revised manuscript and appreciate the reviewer's efforts to improve the quality of our work.

Reviewer #2 (Remarks to the Author):

SUMMARY

The authors have responded inappropriately to criticism throughout. Their study remains of extremely poor design with use of impure commercial methylglyoxal. The authors mention a recent paper by other investigators robustly quantifying methylglyoxal glycation adducts in vitro and in vivo. The finding was that histone proteins have low, ca. 1% modification by major methylglyoxal-derived glycation adduct MG-H1; crosslinks are much lower than this, usually <0.01%. The current study claiming high levels of methylglyoxal crosslinking of histone proteins is therefore lacking physiological relevance because of inappropriate experimental design and implementation. It is presented as otherwise throughout by the authors which is highly misleading.

We respectfully disagree with Reviewer 2's comments raised in this summary:

- (1) Sigma-Aldrich methylglyoxal (product number M0252) has passed rigorous industrial quality control analysis (the corresponding Certificate of Analysis is attached) and been used as it is (with no further purification or chemical analysis) in over 120 publications to date, including the one mentioned by the reviewer (Galligan *et al*, *PNAS*, 2018). Additionally, if any portion of the MGO is hydrated or oxidized, it would be to the inactive lactate or pyruvate species, which would make the effective concentration of MGO used in our experiments even lower than stated. Still, to address the reviewer's comment, we performed a thin-layer chromatography (TLC) analysis that is a standard in separating chemical species (**Figure R1**). **Our analysis clearly indicates that the MGO we used in our experiments has no formaldehyde present.** Finally, as we mentioned in our previous response letter, even if any were present, it would have absolutely no effect on our analyses since the anti-MGO antibody does not recognize such adducts (**Figure R2**).
- (2) The recent illustration of low levels of MGO-glycated histones in healthy tissues and cell lines cultured in low-glucose medium (Galligan *et al*, *PNAS*, 2018, Figure 2) is in complete agreement with our findings that MGO-histone glycation is a pathophysiological phenomenon that we found to occur in high metabolic and damaged cells such as cancer tumors (i.e., in healthy cells DJ-1 prevents this accumulation). We review this conclusion in detail in our discussion under the sub-header: "Histone glycation as a pathophysiological mark".
- (3) In this *PNAS* paper the authors used different analytical methods: (1) for western blot analysis, they used antibodies raised against distinct glycation products (MG-H1, MG-H2 or MG-H3/CEA), which recognize the single glycation product respectively (Galligan *et al*, *PNAS*, 2018, Figure S7) and cannot identify crosslinked products. In addition, the uncropped immunoblots of Figure S7 were not provided in the SI, so we have no way of knowing whether there are high-molecular weight species present; (2) LC-MS/MS analysis, as we mentioned before, has major limitations, one of which is that the crosslinked products of histone glycation cannot be detected due to the limitation of enzymatic digestion and MS database.

- (4) The MGO levels in HEK293 cells were measured at approximately 14 pmol MGO/ng Protein (Galligan *et al*, *PNAS*, 2018, Figure 2). If we assume the averaged molecular weight for a mammalian protein is 50 kD, the calculated molar ratio of MGO to protein is 700:1, which is the highest concentration used in our *in vitro* assays.
- (5) We used exogenous MGO in our cultured cells experiments in order to mimic extreme and long exposures that only exist in diseased states. We have been using those conditions as a setup to study this phenomenon, **just like a cellular enzymatic event would be studied using over-expression or knock out of enzymes** (neither of which is physiological either). The physiological relevance of histone glycation is illustrated by its basal accumulation in breast cancer cell lines, xenografts and patient tumor samples that show similar levels of glycation with no treatment prior to analysis.

Figure R1. TLC analysis of Sigma-Aldrich methylglyoxal (MGO, left lane), formaldehyde (FA, right lane) and their mixture (middle lane). The chromogenic agent used here is KMnO_4 .

Figure R2. Western bolt analysis of H3 treated with either 0.5 mM formaldehyde or 0.5 mM MGO.

BRIEF DETAILED RESPONSES

RESPONSE TO REVIEWER'S COMMENTS

Reviewer 2, point 5 “The guanidino group of arginine residues that participates in glycation is not an amino group.”

AUTHORS' RESPONSE: According to the currently accepted mechanism, the aldehyde group of MGO first reacts with the amino group of arginine and then the MGO ketone group reacts with the arginine guanidino group to form the five member ring glycation product (Scheme 1).

FURTHER COMMENT FROM REVIEWER: Unfortunately the authors do not understand or appreciate delocalisation of charge across the two terminal nitrogen atoms and linking carbon in ac. An amino group does not have this characteristic and hence the guanidino group cannot be described as an amino group. The response is incorrect and an inappropriate response to criticism.

We have revised the statements in the text to now read:

“Glycation is one of the most prevalent NECMs and is characterized by the condensation of the aldehyde form of monosaccharides (such as glucose and fructose) or glycolytic by-products (such as methylglyoxal, MGO) to reactive amino acid residues (mainly primary amines in lysines and guanidino groups in arginines) via the Maillard reaction, forming stable adducts.”

Reviewer 2, point 6. The formation of the major advanced glycation endproduct (AGE) quantitatively, MGO-derived hydroimidazolone MG-H1, and other AGEs are formed non-oxidatively, It should be stated that the major AGEs quantitatively, arginine-derived hydroimidazolones, have slow chemical dynamic reversibility; for example, MG-H1 reverses spontaneously on long-lived proteins with a half-life of about 12 days. So it is not correct to generally state that formation of AGEs cannot be reversed.

AUTHORS' RESPONSE: We apologize for the confusing description of AGEs in the introduction. By definition, advanced glycation endproducts (AGEs) are crosslinked end products rich in aromatic rings or other stable linear structures (e.g., carboxymethyl and carboxyethyl lysine) and cannot undergo any further conversions or be revised by any known enzymes (*Angew Chem. Int. Ed. Engl.*, 2014, 53(39), 10316-10329). Thus, MG-H1 is by definition not an AGE but a glycation intermediate. etc

FURTHER COMMENT FROM REVIEWER: The definition of advanced glycation endproducts (AGEs) as crosslinked end products rich in aromatic rings or other stable linear structures (e.g., carboxymethyl and carboxyethyl lysine) and cannot undergo any further conversions or be revised by any known enzymes is NOT one supported by the article mentioned above by the authors (*Angew Chem. Int. Ed. Engl.*, 2014, 53(39), 10316-10329) NOR generally accepted in the glycation field and beyond. Indeed, in this very article the authors, Hellwig and Henle, describe MG-H1 as an AGE. The authors have presumably read the article and are therefore knowingly providing false evidence with intent to mislead the Editor and reviewers. Clearly, this is an incorrect and an inappropriate response to criticism.

The *Angew Chem.* authors Hellwig and Henle did not provide a complete definition of AGEs in the paper (it is an extensive review paper about the Maillard Reaction). We found the definition of AGEs in other papers:

“Advanced glycation end products (AGEs) are modifications of proteins or lipids that become nonenzymatically glycosylated and oxidized after contact with aldose sugars. Early glycation and oxidation processes result in the

formation of Schiff bases and Amadori products. Further glycation of proteins and lipids causes molecular rearrangements that lead to the generation of AGEs. AGEs may fluoresce, produce reactive oxygen species (ROS), bind to specific cell surface receptors, and form cross-links. AGEs form in vivo in hyperglycemic environments and during aging and contribute to the pathophysiology of vascular disease in diabetes” (Goldin and *et al*, *Circulation*, 2006; Vistoli and *et al*, *Free Radical Research*, 2013).

The original statement in our paper reads:

“The initial glycation adduct can further oxidize and rearrange to form a series of stable products, which can undergo additional chemical transformations including the ability to form crosslinks, yielding species generally referred to as advanced glycation end-products (AGEs)”.

We do not think it is inconsistent or contradictory to either the AGE definition or the reviewer’s comment.

Reviewer 2, point 7. It should be stated that the early attempt to assay AGEs in histone proteins by fluorescence (reference 15), now known to be not specific nor quantitative for AGE content, requires re-assessment. Similarly, the study in reference 16 lacks physiological relevance – using D-glucose 500 mM.

AUTHORS’ RESPONSE: We agree with the reviewer that very little is known about histone glycation and its effect on chromatin structure and function...etc

FURTHER COMMENT FROM THE REVIEWER: The authors did not make the requested comments on the old article cited that used an insecure analytical method for AGE measurement and inappropriate experimental conditions. This an inappropriate response to the criticism.

We cited this paper as an early example for experimental illustration that histones extracted from diabetic mice have increased AGE levels compared to histones extracted from healthy mice. **This was one of the first reports of this phenomenon using a low-resolution measurement and we did not intend to suggest by any means it is quantitative.** To avoid any misunderstandings, we deleted “three-fold” in the original text that now reads:

*“An early **low-resolution** analysis of glycation performed on histones extracted from diabetic mouse liver cells indicated an increase in AGE levels compared to histones extracted from healthy liver cells (15).”*

Reviewer 2, point 8. The SDS-PAGE analysis is indicating oligomerisation – including of peptide backbone fragmentation products (producing the peptides over a continuous and wide range of molecular masses). This is an artefact of the methodology used.

AUTHORS’ RESPONSE: As far as we know, MGO glycation cannot cause the cleavage or fragmentation of target proteins. We do not have a western blot analysis of peptide glycation (or protein glycation post digestion) in our manuscript, so there are no peptide fragmentation adducts in our assays. The laddering we observe in protein and nucleosome glycation is due to a variety of glycation adducts including crosslinking as these species are recognized by both anti MGO and anti H3 (also see the LC-MS analysis in Figure R3, below).

FURTHER COMMENT FROM THE REVIEWER: In Figure 1 the authors show bands with a continuous mass range of polypeptides. This does not reflect oligomers which would rather show discontinuous multi-banded ladders bands with molecular masses of multiples of the molecular mass of the oligomer. The continuous band reflects inappropriate sample processing – heating to high

temperatures during pre-analytic processing, as indicated in the initial review. Rather than correct this technical flaw, the authors have sort to mislead. The small molecular mass increments of glycation adducts (<100 Da) do not produce discernible mass shift on this mass resolving range of polyacrylamide gels. This an inappropriate response to the criticism.

As we clearly indicated in the Materials & Methods section as well as in the first rebuttal letter, we did not boil, heat or lyophilize any of the samples analyzed in this manuscript. In addition, MGO glycation cannot cause the cleavage or fragmentation of any target proteins. There are no “fragments” generated in our glycation assays (please refer to the native PAGE western blot analyses: Figures 2, 4a and S7). The laddering signals we observed in denaturing western blot analysis (SDS-PAGE) are all larger than the monomeric histones, so they cannot be “fragments” of glycated histones.

Regarding “laddering,” the claim that “the small molecular mass increments of glycation adducts (< 100 Da) do not produce discernible mass shift on this mass resolving range of polyacrylamide gels” does not consider the fact that multiple residues are decorated (as indicated in Galliger, *PNAS*, 2018), with adducts ranging from 56-144 Da at minimum (**Figure R3**). In addition, capping of the positive charges of histone by glycation, intramolecular/intermolecular crosslinking and the glycation of multiple types of side chains (Lys, Arg, Cys, His, Pro and *etc*), all will change the histones’ retention on the SDS-PAGE and explain the observed laddering and smearing. Indeed, when we injected the full length MGO-glycated H3 without enzymatic digestion on LC/MS, there is a huge overall mass shift reaching 100 kDa (**Figure R4**).

Standard trypsin digestion-based LC-MS/MS cannot provide any information about crosslinked products. This is also the reason why we are working on developing new mass spec methodologies for analyzing histone glycation in collaboration with histone mass-spectrometry expert Prof. Benjamin Garcia (UPenn). However, the focus of this paper is the qualitative biological consequences of histone glycation *in vitro* and *in vivo*. Thus, we performed a qualitative characterization of MGO-mediated histone glycation instead of developing new quantitative analysis methods.

Figure R3. Structures of MGO-mediated protein glycation products, not including crosslinked products and glycation products on other reactive residues (Cys, His, Pro and *etc*).

Figure R4. LC-MS analysis of MGO-treated full length H3.

Reviewer 2, point 9. In vitro glycation experiments have been performed with MG concentrations at least ≥ 20 -fold higher than found physiologically and with commercial MGO which is known to content high levels of crosslinking agent, formaldehyde [1]. The **AUTHORS' RESPONSE:** MGO solution we used throughout the manuscript was purchased from Sigma-Aldrich (catalog number 67028), is highly pure, and was used in more than 40 peer-reviewed papers. The reference the reviewer mentions (*J. Biol. Chem.*, 1986, 261(30), 14240-14244) did not designate the source or purity of the MGO used in their study, however, we agree this could have been a poor quality MGO as the experiments were performed more than 30 years ago etc.”

FURTHER COMMENT FROM THE REVIEWER: The authors provide a counter claim that the methylglyoxal that they used was highly pure because it was purchased from Sigma Aldrich or “as far as we know”— rather than providing analytical data. This defence of obliviousness may be allowed in a first submission but to return with a revision without addressing the concern is unacceptable. In a recent *Nature Protocol* method on the measurement of methylglyoxal is was accepted by leading experts in peer review that methylglyoxal as supplied by Sigma is highly impure and a method was given for high purity methylglyoxal preparation (*Nature Protocols* 9: 1969-1979, 2014). The authors are being disingenuous and seeking to defend an indefensible position. If they believe the methylglyoxal used herein then they must provide analytical data in support of this. There is no substitute here for repeating the experiments with high purity methylglyoxal.

The authors seek to defence the use of supraphysiological concentrations of methylglyoxal because

“glycation reactions occur over a long period of time”. This fails to observe fundamental concepts in pharmacology of the concentration-response relationship. If the authors are claiming physiological relevance of their research then physiologically relevant concentrations of methylglyoxal must be used. The current studies are only relevant to acute intoxication with very high dose of exogenous methylglyoxal (impure).

As we stated above, Sigma-Aldrich methylglyoxal (product number M0252) has passed rigorous industrial quality control analysis (the corresponding Certificate of Analysis is attached) and been used as it is (with no further purification or chemical analysis) in over 120 publications to date, including the one mentioned by the reviewer (Galligan *et al*, *PNAS*, 2018). Additionally, if any portion of the MGO is hydrated or oxidized, it would be to the inactive lactate or pyruvate species, which would make the effective concentration of MGO used in our experiments even lower than stated. Still, to address the reviewer’s comment, we performed a thin-layer chromatography (TLC) analysis that is a standard in separating chemical species (**Figure R1**). **Our analysis clearly indicates that the MGO we used in our experiments has no formaldehyde present.** Finally, as we mentioned in our previous response letter, even if any were present, it would have absolutely no effect on our analyses since the anti-MGO antibody does not recognize such adducts (**Figure R2**).

The paper the reviewer references several times for detection of MGO impurity (*Nature Protocols* 9: 1969-1979, 2014) states:

“High-purity MG is required to avoid potential contaminating interferences that may affect analyte detection, recovery and calibration. MG from commercial suppliers contains formaldehyde (7 mol% reported⁶⁷) and other impurities. The following method has proven to be effective over >20 years of use in our laboratory³². Attempts to purify commercial MG by removal of impurities are usually less successful.”

However, this paper did not analyze commercialized MGO nor found any artifacts caused by the commercialized MGO. The above statement relies **solely** on **Ref 67**. As we mentioned before, Ref 67 (*J. Biol. Chem.*, 1986, 261(30), 14240-14244) was published more than **30 years ago**. It is thus irrelevant to compare the purity of the current commercial MGO and the one produced 30 years ago. Importantly, reference 67 states that:

“the 40% aqueous solutions of methylglyoxal from Sigma and from Fluka contain ~9 and ~17 mol % formaldehyde, respectively, on the basis of assays using chromotropic acid (12)”.

The colorimetric acid assay reference 67 performed is a UV-dependent method that is very low resolution. It was reported in Reference 12, which is a book published in 1970 (Weiss, F. T., 1970, *Determination of Organic Compounds: Methods and Procedures*, Vol. 32, pp. 102-106, Wiley-Interscience, New York). In fact, the actual colorimetric assay results were not provided in the main text or the SI for us to examine. Sigma-Aldrich uses a far superior and more reliable enzymatic assay for its quality-control analysis (please see the attached Certificate of Analysis).

Reviewer 2, Point 11. Glycation adduct content should be analysed by a robust technique – such as stable isotopic dilution analysis LC-MS/MS after exhaustive enzymatic hydrolysis.

We agree with the reviewer...etc

FURTHER COMMENT FROM THE REVIEWER: The authors agreed with the reviewer and then do no further work to provide robust analytical data. This an inappropriate response to the criticism.

We have clearly stated our reasoning for using western blot analysis over mass spectrometry in our last rebuttal letter:

“The reason we were reluctant to use ms in our analyses are the following key disadvantages in applying it to histone glycation: (1) Sample processing (particularly for histone proteins that are highly rich in lysines and arginines) can tamper the results as it relies on anhydride and ammonia treatment (2) Standard ms is not suitable for analyzing crosslinked protein products due to the limitation of ms fragment database establishment (3) After glycation most of the lysines and arginines are unavailable for trypsin digestion as they are blocked by MGO which generates peptides too long for this analysis; (4) Glycated peptides are less charged and thus do not easily get ionized or “fly” on the instrument (5) It is extremely challenging to identify glycated peptides as the products of MGO-treated proteins are very diverse and need to be specifically targeted. In fact, when we performed low-resolution ms analysis of MGO-treated H3, our results indicated not only the high reactivity of MGO and histones but also how challenging it is to characterize the products by ms. While western blot has its disadvantages, primarily lower sensitivity than ms, we believe in this case the advantages prevail: (1) It is efficient and reliable; (2) The antibodies can be customized to recognize different glycation products in different stages (*Bioorg. Med. Chem. Lett.* 2015, 25(21), 4881-4886) (3) the promiscuous antibody recognizes many sites and forms of MGO-glycation (details of the antibody can be found in our supplementary information).”

The greatest advantage of using LC-MS/MS in this case is to identify the precise sites of modification, however, in our manuscript that is not the key point. As we mentioned above, the aim of this paper is to elucidate the biological consequences of histone glycation and deglycation *in vitro* and *in vivo* and characterize the qualitative effect of MGO-mediated glycation. Requesting that we repeat all the assays in the manuscript using quantitative mass spectrometry analysis (which has critical flaws we detailed above) to also gain precise quantitation of the signal is technically unreasonable and if performed by mass-spectrometry experts, would serve as a completely separate (perhaps complementary) manuscript.

Reviewer 12 - 22. Relating to the use of extremely high, supraphysiological concentrations of methylglyoxal throughout whilst claiming physiological significance. The authors have not performed requested experiments with physiologically methylglyoxal concentrations nor assayed methylglyoxal adduct content of their modified proteins. They cite independent work in a recent paper with robust estimation of MG adduct content in histone proteins (PNAS, 2018, 115(37), 9228-9233

FURTHER COMMENT FROM THE REVIEWER: The attempts to justify use of high supraphysiological concentrations of methylglyoxal by citing previous work that has similar poor design and claiming that methylglyoxal (molecular mass 72 Da) may accumulate in the nucleus and not pass through the nuclear membrane (molecular mass cut-off of at least 30,000 Da) seem, frankly, a desperate attempt to publish than to do research of high quality and impact. The arguments do not stand up to scientific scrutiny. In the paper cited by the authors above on MG-H1 adducts in histone proteins (PNAS, 2018, 115(37), 9228-9233), the glycation adduct content of histone protein *in vitro* and *in vivo* was ca. 0.5 pmol per nmol leu, equivalent to ca. 1% histone modification i.e. 1 in every 100 histone proteins bearing one MG-H1 modification. MG-derived crosslinks are much lower than this - <1% of total methylglyoxal modifications (one modification in every 10,000 histone molecules). The highly oligomerised,

methylglyoxal-modified proteins produced herein lack physiological relevance. To claim otherwise to misleading and deceiving.

As we stated above, the recent illustration of low levels of MGO-glycated histones in healthy tissues and cell lines cultured in low-glucose medium (Galligan *et al*, *PNAS*, 2018, Figure 2) is in complete agreement with our findings that MGO-histone glycation is a pathophysiological phenomenon that we found to occur in high metabolic and damaged cells such as cancer tumors (in normal cells DJ-1 prevents this accumulation). We review this conclusion in detail in our discussion under the sub-header: “Histone glycation as a pathophysiological mark”.

In this *PNAS* paper the authors used different analytical methods: (1) for western blot analysis, they used antibodies raised against distinct glycation products (MG-H1, MG-H2 or MG-H3/CEA), which recognize the single glycation product respectively (Galligan *et al*, *PNAS*, 2018, Figure S7) and cannot specifically identify crosslinked products. In addition, the uncropped immunoblots of Figure S7 were not provided in the SI, so we have no way of knowing whether there are high-molecular species present; (2) LC-MS/MS analysis, as we mentioned before, has major limitations, one of which is that the crosslinked products of histone glycation cannot be detected due to the limitation of enzymatic digestion and MS database.

The MGO levels in HEK293 cells were measured at approximately 14 pmol MGO/ng Protein (Galligan *et al*, *PNAS*, 2018, Figure 2). If we assume the averaged molecular weight for a mammalian protein is 50 kD, the calculated molar ratio of MGO to protein is 700:1, which is the highest concentration used in our *in vitro* assays. Since the report did not include the absolute proportion of histone glycation products at different stages, we are not sure how the reviewer calculated that “histone proteins have low, ca. 1% modification by major methylglyoxal-derived glycation adduct MG-H1; crosslinks are much lower than this, usually <0.01%.”

Reviewer 2, point 23. The way to see an effect of carnosine, of course, is to use high supraphysiological levels of MG and extracellular carnosine – which is done here – to force interaction by experimental design rather than physiologically relevant test. Again, the authors unfortunately use inappropriate experimental design and implementation. These data are unreliable and the study requires repeating with appropriate experimental design and methodology.

AUTHORS’ RESPONSE: We used carnosine in our assay to illustrate that the effect we observe on histones is MGO-dependent. etc

FURTHER COMMENT FROM THE REVIEWER: Carnosine is not a specific scavenger of methylglyoxal and participates in many other types of reactions. Hence, the design is flawed. The original and now additional critique stand. These data are unreliable and the study requires repeating with appropriate experimental design and methodology – using a specific scavenger of methylglyoxal. This is an inappropriate response to the critique. The is an inappropriate and incorrect response to criticism.

While it is true that carnosine is a scavenger for both ROS and MGO, it is nevertheless the most commonly used small molecule scavenger in the study of MGO biology (*J. Cell Mol. Med.* 2011, 15(6): 1339-1354; *eLife*, 2016, 5, pii: e19375; *Sci. Rep.* 2017, 7: 11722; *Biochemical and Biophysical Research Communications*, 1998, 248(1), 28-32; *etc*). Since we used carnosine in similar manner to previous publications (both in terms of purpose, MGO scavenging, as well as concentration and time) in order to show scavenging of endogenous MGO and loss of basal glycation (of breast cancer cells not treated with MGO) as well as decreased sensitivity to MGO, we stand

by our use of carnosine in these experiments.

DISCUSSION

Reviewer 2, point 24. This is based on generally an artefactual experimental data set. I have not read this since it will change profoundly when experiments are repeated with appropriate design and methodology.

AUTHORS' RESPONSE: Our discussion explains our experimental design as well as analyzes our results in light of other publications in the field, which address the reviewer's comments.

FURTHER COMMENT FROM THE REVIEWER: The authors have not addressed the flaws in this work and have sought to defend and describe severely flawed research rather than repeat experimental work with appropriate methodology and design. In other responses they have sought to mislead. The original critique therefore stands.

We believe that reading the manuscript in its entirety, which this reviewer refused to do twice, is part of an adequate and responsible peer review process.

Suggesting that we are "misleading" (without even reading our manuscript in its entirety) is unprofessional, irresponsible and unacceptable.

Reviewer #3 (Remarks to the Author):

The authors have submitted a well revised version of their manuscript. They have answered all of my questions, and have provided additional explanations/clarifications where required. I am happy to accept this revised version as is and have no further remarks. Congratulations on a great story!

We are happy to see that Reviewer 3 is satisfied with our revised manuscript and appreciate the reviewer's efforts to improve the quality of our work.

Certificate of Analysis

Product Name: METHYLGLYOXAL SOLUTION
~ 40 % in water
Product Number: M0252
Batch Number: BCBW8221
Brand: Sigma
CAS Number: 78-98-8
Formula: CH₃COCHO
Formula Weight: 72.06
Storage Temperature: 2-8 C
Quality Release Date: 12 APR 2018

TEST	SPECIFICATION	RESULT
APPEARANCE (COLOR)	YELLOW TO DARK BROWN	DARK BROWN-YELLOW
APPEARANCE (FORM)	LIQUID	LIQUID
ASSAY (ENZYM.)	APPROX. 40 %	34.15 %

Dr. Reinhold Schwenninger
Quality Assurance
Buchs, Switzerland

Sigma-Aldrich warrants that at the time of the quality release or subsequent retest date this product conformed to the information contained in this publication. The current specification sheet may be available at Sigma-Aldrich.com. For further inquiries, please contact Technical Service. Purchaser must determine the suitability of the product for its particular use. See reverse side of invoice or packing slip for additional terms and conditions of sale.